# Retrieval-Augmented Diffusion Models

**Andreas Blattmann**[*]     **Robin Rombach**[*]     **Kaan Oktay**     **Jonas Müller**     **Björn Ommer**

LMU Munich, MCML & IWR, Heidelberg University, Germany

## Abstract

Novel architectures have recently improved generative image synthesis leading to excellent visual quality in various tasks. Much of this success is due to the scalability of these architectures and hence caused by a dramatic increase in model complexity and in the computational resources invested in training these models. Our work[1] questions the underlying paradigm of compressing large training data into ever growing parametric representations. We rather present an orthogonal, semi-parametric approach. We complement comparably small diffusion or autoregressive models with a separate image database and a retrieval strategy. During training we retrieve a set of nearest neighbors from this external database for each training instance and condition the generative model on these informative samples. While the retrieval approach is providing the (local) content, the model is focusing on learning the composition of scenes based on this content. As demonstrated by our experiments, simply swapping the database for one with different contents transfers a trained model post-hoc to a novel domain. The evaluation shows competitive performance on tasks which the generative model has not been trained on, such as class-conditional synthesis, zero-shot stylization or text-to-image synthesis without requiring paired text-image data. With negligible memory and computational overhead for the external database and retrieval we can significantly reduce the parameter count of the generative model and still outperform the state-of-the-art.

## 1   Introduction

Deep generative modeling has made tremendous leaps; especially in language modeling as well as in generative synthesis of high-fidelity images and other data types. In particular for images, astounding results have recently been achieved [22, 15, 56, 59], and three main factors can be identified as the driving forces behind this progress: First, the success of the transformer [88] has caused an architectural revolution in many vision tasks [19], for image synthesis especially through its combination with autoregressive modeling [22, 58]. Second, since their rediscovery, diffusion models have been applied to high-resolution image generation [76, 78, 33] and, within a very short time, set new standards in generative image modeling [15, 34, 63, 59]. Third, these approaches *scale* well [58, 59, 37, 81]; in particular when considering the model- and batch sizes involved for high-quality models [15, 56, 58, 59] there is evidence that this scalability is of central importance for their performance.

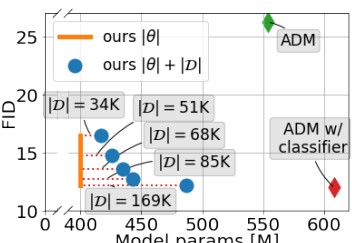

Figure 1: Our semi-parametric model outperforms the unconditional SOTA model ADM [15] on ImageNet [13] and even reaches the class-conditional ADM (ADM w/ classifier), while reducing parameter count. $|\mathcal{D}|$: Number of instances in database at inference; $|\theta|$: Number of trainable parameters.

However, the driving force underlying this training paradigm are models with ever growing numbers of parameters [81] that require huge computational resources. Besides the enormous demands in energy consumption and training time, this paradigm renders future generative modeling more and more exclusive to privileged institutions, thus hindering the democratization of research. Therefore, we here

---

[*]The first two authors contributed equally to this work.

[1]Code is available at `https://github.com/CompVis/retrieval-augmented-diffusion-models`

36th Conference on Neural Information Processing Systems (NeurIPS 2022).

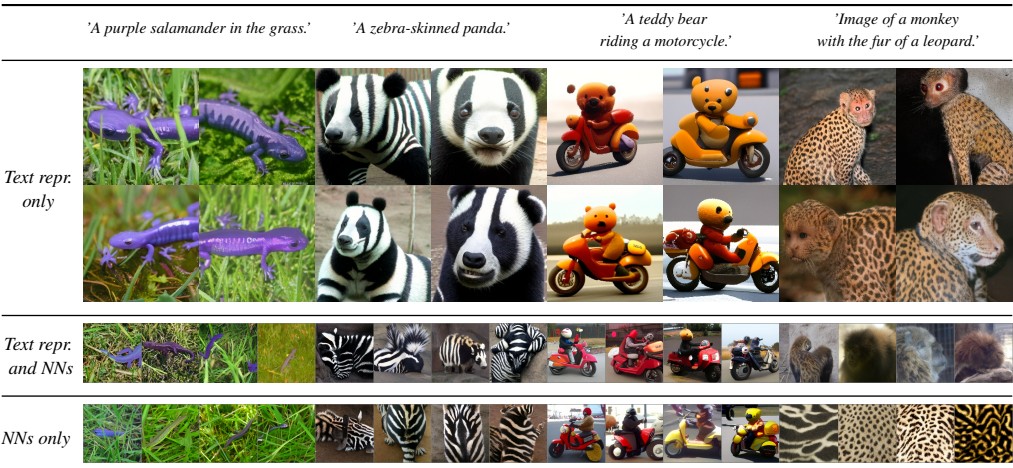

Figure 2: As we retrieve nearest neighbors in the shared text-image space provided by CLIP, we can use text prompts as queries for exemplar-based synthesis. We observe our *RDM* to readily generalize to unseen and fictional text prompts when building the set of retrieved neighbors by directly conditioning on the CLIP text encoding $\phi_{\text{CLIP}}(c_{\text{text}})$ (top row). When using $\phi_{\text{CLIP}}(c_{\text{text}})$ together with its $k-1$ nearest neighbors from the retrieval database (middle row) or the $k$ nearest neighbors alone without the text representation, the model does not show these generalization capabilities (bottom row).

present an orthogonal approach. Inspired by recent advances in retrieval-augmented NLP [4, 89], we question the prevalent approach of expensively compressing visual concepts shared between distinct training examples into large numbers of trainable parameters and equip a comparably small generative model with a large image database. During training, our resulting *semi-parametric* generative models access this database via a nearest neighbor lookup and, thus, need not learn to generate data 'from scratch'. Instead, they learn to *compose* new scenes based on retrieved visual instances. This property not only increases generative performance with reduced parameter count (see Fig. 1), and lowers compute requirements during training. Our proposed approach also enables the models during inference to generalize to new knowledge in form of alternative image databases without requiring further training, what can be interpreted as a form of post-hoc model modification [4]. We show this by replacing the retrieval database with the WikiArt [66] dataset after training, thus applying the model to zero-shot stylization.

Furthermore, our approach is formulated indepently of the underlying generative model, allowing us to present both retrieval-augmented diffusion (*RDM*) and autoregressive (*RARM*) models. By searching in and conditioning on the latent space of CLIP [57] and using scaNN [28] for the NN-search, the retrieval causes negligible overheads in training/inference time (0.95 ms to retrieve 20 nearest neighbors from a database of 20M examples) and storage space (2GB per 1M examples). We show that semi-parametric models yield high fidelity and diverse samples: *RDM* surpasses recent state-of-the-art diffusion models in terms of FID and diversity while requiring less trainable parameters. Furthermore, the shared image-text feature space of CLIP allows for various conditional applications such as text-to-image or class-conditional synthesis, despite being trained on images only (as demonstrated in Fig. 2). Finally, we present additional truncation strategies to control the synthesis process which can be combined with model specific sampling techniques such as classifier-free guidance for diffusion models [32] or top-$k$ sampling [23] for autoregressive models.

## 2   Related Work

**Generative Models for Image Synthesis.** Generating high quality novel images has long been a challenge for deep learning community due to their high dimensional nature. Generative adversarial networks (GANs) [25] excel at synthesizing such high resolution images with outstanding quality [5, 39, 40, 70] while optimizing their training objective requires some sort of tricks [1, 27, 54, 53] and their samples suffer from the lack of diversity [80, 1, 55, 50]. On the contrary, likelihood-based methods have better training properties and they are easier to optimize thanks to their ability to capture the full data distribution. While failing to achieve the image fidelity of GANs, variational autoencoders (VAEs) [43, 61] and flow-based methods [16, 17] facilitate high resolution image generation with fast sampling speed [84, 45]. Autoregressive models (ARMs) [10, 85, 87, 68] succeed in density estimation like the other likelihood-based methods, albeit at the expense of computational efficiency. Starting with the seminal works of Sohl-Dickstein et al. [76] and Ho et

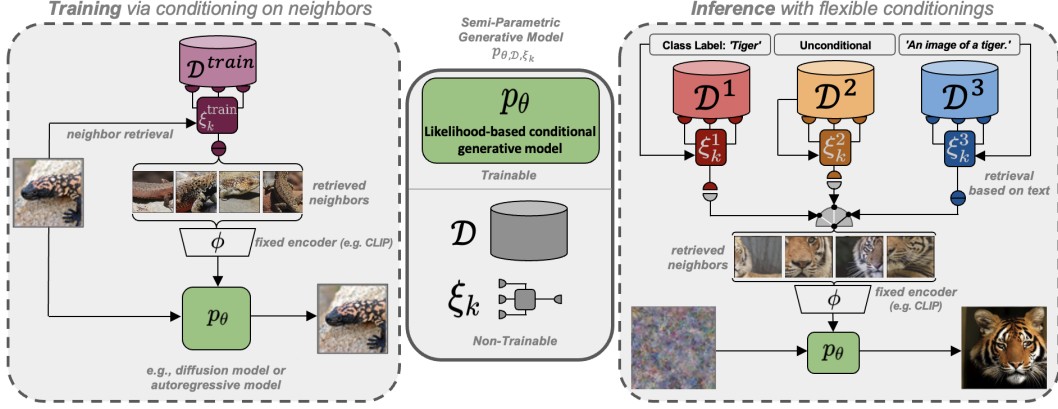

Figure 3: A semi-parametric generative model consists of a trainable conditional generative model (decoding head) $p_\theta(x|\cdot)$, an external database $\mathcal{D}$ containing visual examples and a sampling strategy $\xi_k$ to obtain a subset $\mathcal{M}_\mathcal{D}^{(k)} \subseteq \mathcal{D}$, which serves as conditioning for $p_\theta$. During training, $\xi_k$ retrieves the nearest neighbors of each target example from $\mathcal{D}$, such that $p_\theta$ only needs to learn to compose consistent scenes based on $\mathcal{M}_\mathcal{D}^{(k)}$, *cf.* Sec 3.2. During inference, we can exchange $\mathcal{D}$ and $\xi_k$, thus resulting in flexible sampling capabilities such as post-hoc conditioning on class labels ($\xi_k^1$) or text prompts ($\xi_k^3$), *cf.* Sec. 3.3, and zero-shot stylization, *cf.* Sec. 4.3.

al. [33], diffusion-based generative models have improved generative modeling of artificial visual systems [15, 44, 90, 35, 92, 65]. Their good performance, however, comes at the expense of high training costs and slow sampling. To circumvent the drawbacks of ARMs and diffusion models, several two-stage models are proposed to scale them to higher resolutions by training them on the compressed image features [86, 60, 22, 93, 63, 75, 21]. However, they still require large models and significant compute resources, especially for unconditional image generation [15] on complex datasets like ImageNet [13] or complex conditional tasks such as text-to-image generation [56, 58, 26, 63]. To address these issues, given limited compute resources, we propose to trade trainable parameters for an external memory which empowers smaller models to achieve high fidelity image generation.

**Retrieval-Augmented Generative Models.** Using external memory to augment traditional models has recently drawn attention in natural language processing (NLP) [41, 42, 52, 29]. For example, RETRO [4] proposes a retrieval-enhanced transformer for language modeling which performs on par with state-of-the-art models [6] using significantly less parameters and compute resources. These retrieval-augmented models with external memory turn purely parametric deep learning models into semi-parametric ones. Early attempts [51, 74, 83, 91] in retrieval-augmented visual models do not use an external memory and exploit the training data itself for retrieval. In image synthesis, IC-GAN [8] utilizes the neighborhood of training images to train a GAN and generates samples by conditioning on single instances from the training data. However, using training data itself for retrieval potentially limits the generalization capacity, and thus, we favor an external memory in this work.

# 3 Image Synthesis with Retrieval-Augmented Generative Models

Our work considers data points as an explicit *part of the model*. In contrast to common neural generative approaches for image synthesis [5, 40, 70, 60, 22, 10, 9], this approach is not only parameterized by the learnable weights of a neural network, but also a (fixed) set of data representations and a non-learnable *retrieval* function, which, given a query from the training data, retrieves suitable data representations from the external dataset. Following prior work in natural language modeling [4], we implement this retrieval pipeline as a nearest neighbor lookup.

Sec. 3.1 and Sec. 3.2 formalize this approach for training retrieval-augmented diffusion and autoregressive models for image synthesis, while Sec. 3.3 introduces sampling mechanisms that become available once such a model is trained. Fig. 3 provides an overview over our approach.

## 3.1 Retrieval-Enhanced Generative Models of Images

Unlike common, fully parametric neural generative approaches for images, we define a *semi-parametric* generative image model $p_{\theta,\mathcal{D},\xi_k}(x)$ by introducing trainable parameters $\theta$ *and* non-trainable model components $\mathcal{D}, \xi_k$, where $\mathcal{D} = \{y_i\}_{i=1}^N$ is a *fixed* database of images $y_i \in \mathbb{R}^{H_\mathcal{D} \times W_\mathcal{D} \times 3}$ that is disjoint from our train data $\mathcal{X}$. Further, $\xi_k$ denotes a (non-trainable) sampling strategy to obtain a subset of $\mathcal{D}$ based on a query $x$, i.e. $\xi_k \colon x, \mathcal{D} \mapsto \mathcal{M}_\mathcal{D}^{(k)}$, where $\mathcal{M}_\mathcal{D}^{(k)} \subseteq \mathcal{D}$ and $|\mathcal{M}_\mathcal{D}^{(k)}| = k$ . Thus, only $\theta$ is actually learned during training.

Importantly, $\xi_k(x, \mathcal{D})$ has to be chosen such that it provides the model with beneficial visual representations from $\mathcal{D}$ for modeling $x$ and the entire capacity of $\theta$ can be leveraged to *compose* consistent scenes based on these patterns. For instance, considering query images $x \in \mathbb{R}^{H_x \times W_x \times 3}$, a valid strategy $\xi_k(x, \mathcal{D})$ is a function that for each $x$ returns the set of its $k$ nearest neighbors, measured by a given distance function $d(x, \cdot)$.

Next, we propose to provide this retrieved information to the model via *conditioning*, i.e. we specify a general semi-parametric generative model as

$$p_{\theta, \mathcal{D}, \xi_k}(x) = p_\theta(x \mid \xi_k(x, \mathcal{D})) = p_\theta(x \mid \mathcal{M}_{\mathcal{D}}^{(k)}) \tag{1}$$

In principle, one could directly use image samples $y \in \mathcal{M}_{\mathcal{D}}^{(k)}$ to learn $\theta$. However, since images contain many ambiguities and their high dimensionality involves considerable computational and storage cost[2] we use a *fixed*, pre-trained image encoder $\phi$ to project all examples from $\mathcal{M}_{\mathcal{D}}^{(k)}$ onto a low-dimensional manifold. Hence, Eq. (1) reads

$$p_{\theta, \mathcal{D}, \xi_k}(x) = p_\theta(x \mid \{ \phi(y) \mid y \in \xi_k(x, \mathcal{D}) \}). \tag{2}$$

where $p_\theta(x|\cdot)$ is a conditional generative model with trainable parameters $\theta$ which we refer to as *decoding head*. With this, the above procedure can be applied to any type of generative decoding head and is not dependent on its concrete training procedure.

## 3.2 Instances of Semi-Parametric Generative Image Models

During training we are given a train dataset $\mathcal{X} = \{x_i\}_{i=1}^M$ of images whose distribution $p(x)$ we want to approximate with $p_{\theta, \mathcal{D}, \xi_k}(x)$. Our train-time sampling strategy $\xi_k$ uses a query example $x \sim p(x)$ to retrieve its $k$ nearest neighbors $y \in \mathcal{D}$ by implementing $d(x, y)$ as the cosine similarity in the image feature space of CLIP [57]. Given a sufficiently large database $\mathcal{D}$, this strategy ensures that the set of neighbors $\xi_k(x, \mathcal{D})$ shares sufficient information with $x$ and, thus, provides useful visual information for the generative task. We choose CLIP to implement $\xi_k$, because it embeds images in a low dimensional space ($\dim = 512$) and maps semantically similar samples to the same neighborhood, yielding an efficient search space. Fig. 4 visualizes examples of nearest neighbors retrieved via a ViT-B/32 vision transformer [19] backbone.

| $x$ | $\xi_{15}(x)$ |
|---|---|

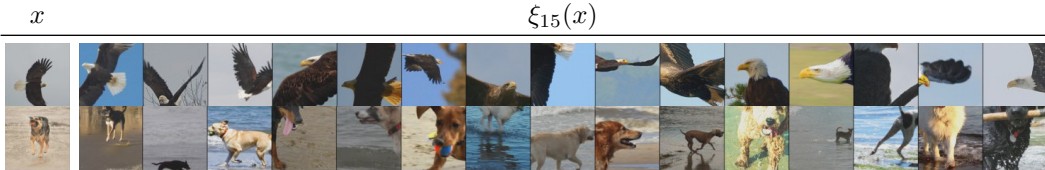

Figure 4: $k = 15$ nearest neighbors from $\mathcal{D}$ for a given query $x$ when parameterizing $d(x, \cdot)$ with CLIP [57].

Note that this approach can, in principle, turn any generative model into a semi-parametric model in the sense of Eq. (2). In this work we focus on models where the decoding head is either implemented as a diffusion or an autoregressive model, motivated by the success of these models in image synthesis [33, 15, 63, 56, 58, 22].

To obtain the image representations via $\phi$, different encoding models are conceivable in principle. Again, the latent space of CLIP offers some advantages since it is (i) very compact, which (ii) also reduces memory requirements. Moreover, the contrastive pretraining objective (iii) provides a shared space of image and text representations, which is beneficial for text-image synthesis, as we show in Sec. 4.2. Unless otherwise specified, $\phi \equiv \phi_{\text{CLIP}}$ is set in the following. We investigate alternative parameterizations of $\phi$ in Sec. E.2.

Note that with this choice, the additional database $\mathcal{D}$ can also be interpreted as a fixed *embedding layer*[3] of dimensionality $|\mathcal{D}| \times 512$ from which the nearest neighbors are retrieved.

### 3.2.1 Retrieval-Augmented Diffusion Models

In order to reduce computational complexity and memory requirements during training, we follow [63] and build on latent diffusion models (LDMs) which learn the data distribution in the latent space $z = E(x)$ of a pretrained autoencoder. We dub this retrieval-augmented latent diffusion model *RDM* and train it with the usual reweighted likelihood objective [33], yielding the objective [76, 33]

$$\min_\theta \mathcal{L} = \mathbb{E}_{p(x), z \sim E(x), \epsilon \sim \mathcal{N}(0,1), t} \left[ \| \epsilon - \epsilon_\theta(z_t, t, \{\phi_{\text{CLIP}}(y) \mid y \in \xi_k(x, \mathcal{D})\}) \|_2^2 \right], \tag{3}$$

---

[2]Note that $\mathcal{D}$ is essentially a part of the model weights

[3]For a database of 1M images and using 32-bit precision, this equals approximately 2.048 GB

where the expectation is approximated by the empirical mean over training examples. In the above equation, $\epsilon_\theta$ denotes the UNet-based [64] denoising autoencoder as used in [15, 63] and $t \sim$ Uniform$\{1, \ldots, T\}$ denotes the time step [76, 33]. To feed the set of nearest neighbor encodings $\phi_{\text{CLIP}}(y)$ into $\epsilon_\theta$, we use the cross-attention conditioning mechanism proposed in [63].

### 3.2.2 Retrieval-Augmented Autoregressive Models

Our approach is applicable to several types of likelihood-based methods. We show this by augmenting diffusion models (Sec. 3.2.1) as well as autoregressive models with the retrieved representations. To implement the latter, we follow [22] and train autoregressive transformer models to model the distribution of the discrete image tokens $z_q = E(x)$ of a VQGAN [22, 86]. Specifically, as for *RDM*, we train retrieval-augmented autoregressive models *(RARMs)* conditioned on the CLIP embeddings $\phi_{\text{CLIP}}(y)$ of the neighbors $y$, so that the objective reads

$$\min_\theta \mathcal{L} = -\mathbb{E}_{p(x), z_q \sim E(x)} \Big[ \sum_i \log p(z_q^{(i)} \mid z_q^{(<i)}, \{\phi_{\text{CLIP}}(y) \mid y \in \xi_k(x, \mathcal{D})\}) \Big] , \qquad (4)$$

where we choose a row-major ordering for the autoregressive factorization of the latent $z_q$. We condition the model on the set of neighbor embeddings $\phi_{\text{CLIP}}(\xi_k(x, \mathcal{D}))$ via cross-attention [88].

## 3.3 Inference for Retrieval-Augmented Generative Models

**Conditional Synthesis without Conditional Training**   Being able to change the (non-learned) $\mathcal{D}$ and $\xi_k$ at test time offers additional flexibility compared to standard generative approaches: Depending on the application, it is possible to extent/restrict $\mathcal{D}$ for particular exemplars; or to skip the retrieval via $\xi_k$ altogether and provide a set of representations $\{\phi_{\text{CLIP}}(y_i)\}_{i=1}^k$ directly. This allows us to use additional conditional information such as a text prompt or a class label, which has not been available during training, to achieve more fine-grained control during synthesis.

For **text-to-image generation**, for example, our model can be conditioned in several ways: Given a text prompt $c_{\text{text}}$ and using the text-to-image retrieval ability of CLIP, we can retrieve $k$ neighbors from $\mathcal{D}$ and use these as an implicit text-based conditioning. However, since we condition on CLIP representations $\phi_{\text{CLIP}}$, we can also condition directly on the *text* embeddings obtained via CLIP's language backbone (since CLIP's text-image embedding space is shared). Accordingly, it is also possible to combine these approaches and use text and image representations simultaneously. We show and compare the results of using these sampling techniques in Fig. 2.

Given a class label $c$, we define a text such as *'An image of a $t(c)$.'* based on its textual description $t(c)$ or apply the embedding strategy for text prompts and sample a pool $\xi_l(c)$, $k \le l$ for each class. By randomly selecting $k$ adjacent examples from this pool for a given query $c$, we obtain an inference-time class-conditional model and analyze these post-hoc conditioning methods in Sec. 4.2.

For **unconditional generative modeling**, we randomly sample a pseudo-query $\tilde{x} \in \mathcal{D}$ to obtain the set $\xi_k^{\text{test}}(\tilde{x}, \mathcal{D})$ of its $k$ nearest neighbors. Given this set, Eq. (2) can be used to draw samples, since $p_\theta(x|\cdot)$ itself is a generative model. However, when generating all samples from $p_{\theta, \mathcal{D}, \xi_k}(x)$ only from one particular set $\xi_k^{\text{test}}(\tilde{x})$, we expect $p_{\theta, \mathcal{D}, \xi_k}(x)$ to be unimodal and sharply peaked around $\tilde{x}$. When intending to model a complex multimodal distribution $p(x)$ of natural images, this choice would obviously lead to weak results. Therefore, we construct a proposal distribution based on $\mathcal{D}$ where

$$p_{\mathcal{D}}(\tilde{x}) = \frac{|\{x \in \mathcal{X} \mid \tilde{x} \in \xi_k(x, \mathcal{D})\}|}{k \cdot |\mathcal{X}|} , \quad \text{for } \tilde{x} \in \mathcal{D} . \qquad (5)$$

This definition counts the instances in the database $\mathcal{D}$ which are useful for modeling the training dataset $\mathcal{X}$. Note that $p_{\mathcal{D}}(\tilde{x})$ only depends on $\mathcal{X}$ and $\mathcal{D}$, what allows us to precompute it. Given $p_{\mathcal{D}}(\tilde{x})$, we can obtain a set

$$\mathcal{P} = \Big\{ x \sim p_\theta(x \mid \{\phi(y) \mid y \in \xi_k(\tilde{x}, \mathcal{D})\}) \Big| \tilde{x} \sim p_{\mathcal{D}}(\tilde{x}) \Big\} \qquad (6)$$

of samples from the our model. We can thus draw from the unconditional modeled density $p_{\theta, \mathcal{D}, \xi_k}(x)$ by drawing $x \sim \text{Uniform}(\mathcal{P})$.

By choosing only a fraction $m \in (0, 1]$ of most likely examples $\tilde{x} \sim p_{\mathcal{D}}(\tilde{x})$, we can artificially truncate this distribution and trade sample quality for diversity. See Sec. D.1. for a detailed description of this mechanism which we call *top-m sampling* and Sec. 4.5 for an empirical demonstration.

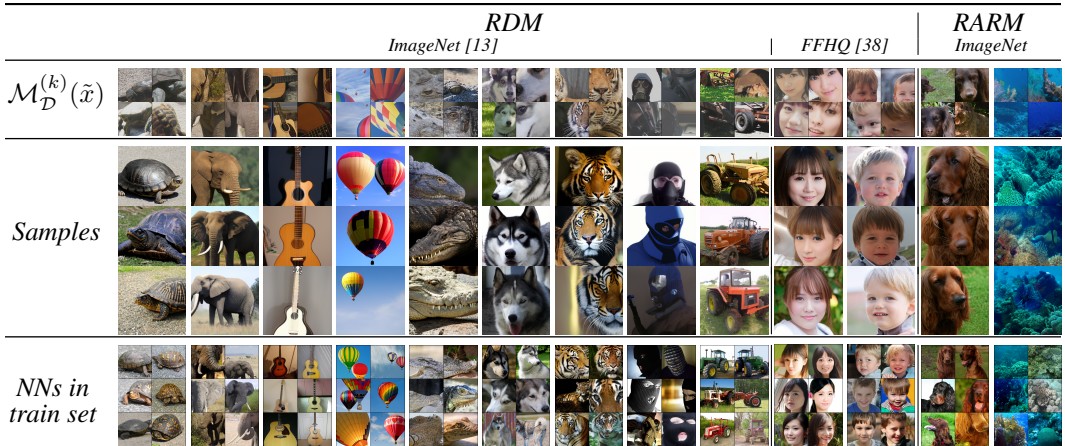

Figure 5: Samples from our unconditional models together with the sets of $\mathcal{M}_\mathcal{D}^{(k)}(\tilde{x})$ of retrieved neighbors for the pseudo query $\tilde{x}$, *cf.* Sec. 3.3, and nearest neighbors from the train set, measured in CLIP [57] feature space. For ImageNet samples are generated with $m = 0.01$, guidance with $s = 2.0$ and 100 DDIM steps for *RDM* and $m = 0.05$, guidance scale $s = 3.0$ and top-$k = 2048$ for *RARM*. On FFHQ we use $s = 1.0$, $m = 0.1$.

## 4 Experiments

This section presents experiments for both retrieval-augmented diffusion and autoregressive models. To obtain nearest neighbors we apply the ScaNN search algorithm [28] in the feature space of a pretrained CLIP-ViT-B/32 [57]. Using this setting, retrieving 20 nearest neighbors from the database described above takes $\sim 0.95$ ms. For more details on our retrieval implementation, see Sec. F.1. For quantitative performance measures we use FID [31], CLIP-FID [48], Inception Score (IS) [67] and Precision-Recall [47], and, for the diffusion models, generate samples with the DDIM sampler [77] with 100 steps and $\eta = 1$. For hyperparameters, implementation and evaluation details *cf.* Sec. F.

### 4.1 Semi-Parametric Image Generation

Drawing pseudo-queries from the proposal distribution proposed in Sec. 3.3 and Eq. (6) enables semi-parametric unconditional image generation. However, before the actual application, we compare different choices of the database $\mathcal{D}_{\text{train}}$ used during training and determine an appropriate choice for the value $k$ of the retrieved neighbors during training.

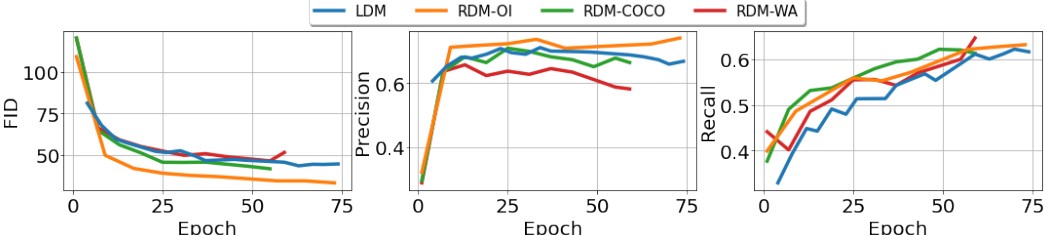

Figure 6: Comparing performance metrics of *RDMs* with different train databases $\mathcal{D}_{\text{train}}$ with those of an *LDM* baseline on the dogs-subset of ImageNet [13]; we find that having a database of diverse visual instance from visual domains similar to the train dataset $\mathcal{X}$ (as *RDM -COCO*) improves performance upon fully-parametric baseline. Increasing the size of the database further boosts performance, leading to significant improvements of *RDMs* over the baseline despite having less trainable parameters.

**Finding a train-time database $\mathcal{D}_{\text{train}}$.** Key to a successful application of semi-parametric models is choosing an appropriate train database $\mathcal{D}_{\text{train}}$, as it has to provide the generative backbone $p_\theta$ with useful information. We hypothesize that a large database with diverse visual instances is most useful for the model, since the probability of finding nearby neighbors in $\mathcal{D}_{\text{train}}$ for *every* train example is highest for this choice. To verify this claim, we compare the visual quality and sample diversity of three *RDMs* trained on the dogs-subset of ImageNet [13] with i) WikiArt [66] (*RDM-WA*), ii) MS-COCO [7] (*RDM-COCO*) and iii) 20M examples obtained by cropping images (see App. F.1) from OpenImages [46] (*RDM-OI*) as train database $\mathcal{D}_{\text{train}}$ with that of an *LDM* baseline with $1.3\times$ more parameters. Fig 6 shows that i) a database $\mathcal{D}_{\text{train}}$, whose examples are from a different domain than those of the train set $\mathcal{X}$ leads to degraded sample quality, whereas ii) a small database from the same domain as $\mathcal{X}$ improves performance compared to the *LDM* baseline. Finally, iii) increasing

the size of $\mathcal{D}_{\text{train}}$ further boosts performance in quality and diversity metrics and leads to significant improvements of *RDMs* compared to *LDMs*.

| Method | FID↓ train | FID↓ val | CLIP-FID↓ train | CLIP-FID↓ val | Precision↑ | Recall↑ |
|---|---|---|---|---|---|---|
| *RDM-IN* | 5.91 | 5.32 | 3.92 | 4.44 | 0.74 | 0.51 |
| *RDM-OI* | 12.28 | 11.31 | 4.09 | 4.59 | 0.69 | 0.55 |
| *RDM-IN/OI* | 17.23 | 16.82 | 8.86 | 9.75 | 0.52 | 0.60 |
| *RDM-OI/IN* | 10.81 | 12.01 | 3.84 | 4.41 | 0.81 | 0.39 |

| Method | FID↓ | CLIP-FID↓ | CLIP-score↑ | IS↑ |
|---|---|---|---|---|
| LAFITE [94] | 26.94 | - | - | **26.02** |
| *RDM-IN* | 27.28 | 18.12 | 0.29 | 24.17 |
| *RDM-OI* | **22.08** | **13.16** | **0.30** | 24.31 |

Table 1: *Generalization to new databases.* Left: We train *RDMs* on ImageNet with OpenImages (*RDM-OI*) and the train dataset itself (*RDM-IN*). By exchanging the train and inference databases between the two models we see that *RDM-OI* which is trained with a database disjoint from the train set generalizes better to new inference databases. Right: Quantitative comparison against LAFITE [94] on zero-shot text-to-image synthesis.

For the above experiment we used $\mathcal{D}_{\text{train}} \cap \mathcal{X} = \emptyset$. This is in contrast to prior work [8] which conditions a generative model on the train dataset itself, i.e., $\mathcal{D}_{\text{train}} = \mathcal{X}$. Our choice is motivated by the aim to obtain a model as general as possible which can be used for more than one task during inference, as introduced in Sec. 3.3. To show the benefits of using $\mathcal{D}_{\text{train}} \cap \mathcal{X} = \emptyset$ we use ImageNet [13] as train set $\mathcal{X}$ and compare *RDM-OI* with an *RDM* conditioned on $\mathcal{X}$ itself (*RDM-IN*). We evaluate their performance on the ImageNet train- and validation-sets in Tab. 1, which shows *RDM-OI* to closely reach the performance of *RDM-IN* in CLIP-FID [48] and achieve more diverse results. When interchanging the test-time database between the two models, i.e., conditioning *RDM-OI* on examples from ImageNet (*RDM-OI/IN*) and vice versa (*RDM-IN/OI*) we observe strong performance degradation of the latter model, whereas the former improves in most metrics and outperforms *RDM-IN* in CLIP-FID, thus showing the enhanced generalization capabilities when choosing $\mathcal{D}_{\text{train}} \cap \mathcal{X} = \emptyset$. To provide further evidence of this property we additionally evaluate the models on zero-shot text-conditional on the COCO dataset [7] in Tab. 1. Again, we observe better image quality (FID) as well as image-text alignment (CLIP-score) of *RDM-OI* which furthermore outperforms LAFITE [94] in FID, despite being trained on only a third of the train examples.

**How many neighbors to retrieve during training?** As the number $k_{\text{train}}$ of retrieved nearest neighbors during training has a strong influence on the properties of the resulting model after training, we first identify hyperparameters obtain a model with optimal synthesis properties. Hence, we parameterize $p_\theta$ with a diffusion model and train five models for

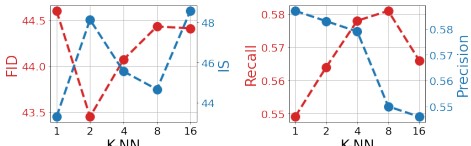

Figure 7: Effect of $k_{\text{train}}$.

different $k_{\text{train}} \in \{1, 2, 4, 8, 16\}$ on ImageNet [13]. All models use identical generative backbones and computational resources (details in Sec. F.2.1). Fig. 7 shows resulting performance metrics assessed on 1000 samples. For FID and IS we do not observe significant trends. Considering precision and recall, however, we see that increasing $k_{\text{train}}$ trades consistency for diversity. Large $k_{\text{train}}$ causes recall, i.e. sample diversity, to deteriorate again.

We attribute this to a regularizing influence of non-redundant, additional information beyond the single nearest neighbor, which is fed to the respective model during training, when $k_{\text{train}} > 1$. For $k_{\text{train}} \in \{2, 4, 8\}$ this additional information is beneficial and the corresponding models appropriately mediate between quality and diversity. Thus, we use $k = 4$ for our main *RDM*. Furthermore, the numbers of neighbors has a significant effect on the generalization capabilities of our model for *conditional* synthesis, *e.g.* text-to-image synthesis as in Fig. 2. We provide an in-depth evaluation of this effect in Sec. 4.2 and conduct a similar study for *RARM* in Sec. E.4.

**Qualitative results.** Fig. 5 shows samples of *RDM*/*RARM* trained on ImageNet as well as *RDM* samples on FFHQ [38] for different sets $\mathcal{M}_{\mathcal{D}}^{(k)}(\tilde{x})$ of retrieved neighbors given a pseudo-query $\tilde{x} \sim p_{\mathcal{D}}(\tilde{x})$. We also plot the nearest neighbors from the train set to show that this set is disjoint from the database $\mathcal{D}$ and that our model renders new, unseen samples.

**Quantitative results.** Tab. 2 compares our model with the recent state-of-the-art diffusion model ADM [15] and the semi-parametric GAN-based model IC-GAN [8] (which requires access to the *training set* examples during inference) in unconditional image synthesis on ImageNet [13] $256 \times 256$.

To boost performance, we use the sampling strategies proposed in Sec. 3.3 (which is also further detailed in Sec. D.1). With classifier-free guidance (c.f.g.), our model attains better scores than IC-GAN and ADM while being on par with ADM-G [15]. The latter requires an additional classifier and the labels of training instances during inference. Without any additional information about training data, e.g., image labels, *RDM* achieves the best overall performance.

| Method | FID↓ | | IS↑ | Precision↑ | | Recall↑ | | $N_{\text{params}}$ | |
|---|---|---|---|---|---|---|---|---|---|
| | train | val | | train | val | train | val | | |
| IC-GAN [8] | 18.17 | 15.60* | 59.00* | 0.77 | **0.73** | 0.21 | 0.23 | 191M | conditioned on train set, add. aug. |
| ADM [15] | 26.21 | 32.50* | 39.70 | 0.61 | - | 0.63 | - | 554M | 250 steps |
| ADM-G [15] | 33.03 | - | 32.92 | 0.56 | - | **0.65** | - | 618M | 250 steps, c.g., s=1.0 |
| ADM-G [15] | 12.00 | - | 95.41 | 0.76 | - | 0.44 | - | 618M | 250 steps, c.g., s=10.0 |
| *RDM-OI* (ours) | 24.50 | 21.28 | 45.29 | 0.60 | 0.54 | **0.65** | **0.66** | 400M | 100 steps, $m = 0.1$ |
| *RDM-OI* (ours) | 19.08 | 16.89 | 62.78 | 0.57 | 0.62 | 0.56 | 0.57 | 400M | 100 steps, $m = 0.01$ |
| *RDM-OI* (ours) | 13.22 | 12.29 | 70.64 | 0.72 | 0.65 | 0.56 | 0.51 | 400M | 100 steps, c.f.g., $s = 1.75$, $m = 0.1$ |
| *RDM-OI* (ours) | 13.60 | 13.11 | 87.58 | **0.79** | **0.73** | 0.51 | 0.50 | 400M | 100 steps, c.f.g., $s = 1.5$, $m = 0.02$ |
| *RDM-OI* (ours) | 12.21 | 11.31 | 77.93 | 0.75 | 0.69 | 0.55 | 0.55 | 400M | 100 steps, c.f.g., $s = 1.5$, $m = 0.05$ |
| *RDM-IN* (ours) | 5.91 | 5.32 | 158.76 | 0.74 | 0.74 | 0.51 | 0.53 | 400M | 100 steps, c.f.g., $s = 1.5$, $m = 0.05$ |

Table 2: Comparison of *RDM* with recent state-of-the-art methods for unconditional image generation on ImageNet [13]. While *c.f.g.* denotes classifier-free guidance with a scale parameter $s$ as proposed in [32], *c.g.* refers to classifier guidance [15], what requires a classifier pretrained on the noisy representations of diffusion models to be available. *: numbers taken from [8].

For $m = 0.1$, our retrieval-augmented diffusion model surpasses unconditional ADM for FID, IS, precision and, without guidance, for recall. For $s = 1.75$, we observe bisected FID scores compared to our unguided model and even reach the guided model ADM-G, which, unlike *RDM*, requires a classifier that is pre-trained on noisy data representations. The optimal parameters for FID are $m = 0.05$, $s = 1.5$, as in the bottom row of Tab. 2. Using

| Method | CLIP-FID | CLIP-Prec | CLIP-Rec |
|---|---|---|---|
| P-GAN [69] | 4.87 | - | - |
| Style-GAN2 [39] | 2.90 | - | - |
| LDM [63] | 2.12 | 0.81 | **0.48** |
| LDM (equal $N_{\text{params}}$) | 2.63 | 0.87 | 0.44 |
| *RDM-OI* | **1.92** | **0.93** | 0.35 |

Table 3: Quantiative results on FFHQ [38]. *RDM-OI* samples generated with $m = 0.1$ and without classifier-free guidance.

these parameters for *RDM-IN* results in a model which even achieves similar FID scores than state of the class-conditional models on ImageNet [63, 15, 70] without requiring any labels during training or inference. Overall, this shows the strong performance of *RDM* and the flexibility of top-m sampling and c.f.g., which we further analyze in Sec. 4.5. Moreover we train an exact replicate of our ImageNet *RDM-OI* on the FFHQ [38] and summarize the results in Tab. 3. Since FID [31] has been shown to be "insensitive to the facial region" [48] we again use CLIP-based metrics. Even for this simple dataset, our retrieval-based strategy proves beneficial, outperforming strong GAN and diffusion baselines, albeit at the cost of lower diversity (recall).

## 4.2 Conditional Synthesis without Conditional Training

**Text-to-Image Synthesis** In Fig. 2, we show the zero-shot text-to-image synthesis capabilities of our ImageNet model for user defined text prompts. When building the set $\mathcal{M}_{\mathcal{D}}^{(k)}(c_{\text{text}})$ by directly using *i)* the CLIP encodings $\phi_{\text{CLIP}}(c_{\text{text}})$ of the actual textual description itself (top row), we interestingly see that our model generalizes to generating fictional descriptions and transfers attributions across object classes. However, when using *ii)* $\phi_{\text{CLIP}}(c_{\text{text}})$ together with its $k - 1$ nearest neighbors from the database $\mathcal{D}$ as done in [2], the model does not generalize to these difficult conditional inputs (mid row). When *iii)* only using the $k$ CLIP image representations of the nearest neighbors, the results are even

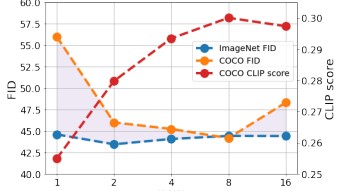

Figure 8: We observe that the number of neighbors $k_{\text{train}}$ retrieved during training significantly impacts the generalization abilities of *RDM*. See Sec. 4.2.

worse (bottom row). We evaluate the text-to-image capabilities of *RDMs* on 30000 examples from the COCO validation set and compare with LAFITE [94]. The latter is also based on CLIP space, but unlike our method, the image features are translated to text features by utilizing a supervised model in order to address the mismatch between CLIP text and image features. Tab. 1 summarizes the results and shows that our *RDM-OI* obtains better image quality as measured by the FID score.

Similar to Sec. 4.1 we investigate the influence of $k_{\text{train}}$ on the text-to-image generalization capability of *RDM*. To this end we evaluate the zero-shot transferability of the ImageNet models presented in the last section to text-conditional image generation and, using strategy i) from the last paragraph, evaluate their performance on 2000 captions from the validation set of COCO [7]. Fig. 8 compares the resulting FID and CLIP scores on COCO for the different choices of $k_{\text{train}}$. As a reference for the train performance, we furthermore plot the ImageNet FID. Similar to Fig. 7 we find that small $k_{\text{train}}$ lead to weak generalization properties, since the corresponding models cannot handle misalignments between the text representation received during inference and image representations it is trained on. Increasing $k_{\text{train}}$ results in sets $\mathcal{M}_{\mathcal{D}}^{(k)}(x)$ which cover a larger feature space volume, what regularizes the corresponding models to be more robust against such misalignments. Consequently, the generalization abilities increase with $k_{\text{train}}$ and reach an optimum at $k_{\text{train}} = 8$. Further increasing $k_{\text{train}}$ results in decreased information provided via the retrieved neighbors (*cf.* Fig. 4) and causes deteriorating generalization capabilities.

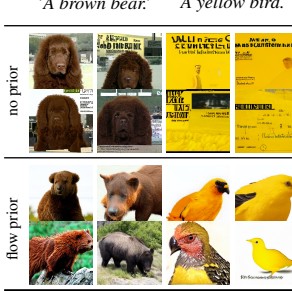

'A brown bear.'     'A yellow bird.'

We note the similarity of this approach to [59], which, by directly conditioning on the CLIP image representations of the data, essentially learns to invert the abstract image embedding. In our framework, this corresponds to $\xi_k(x) = \phi_{\text{CLIP}}(x)$ (i.e., no external database is provided). In order to fix the misalignment between text embeddings and image embeddings, [59] learns a conditional diffusion model for the generative mapping between these representations, requiring paired data. We argue that our retrieval-augmented approach provides an orthogonal approach to this task *without* requiring paired data. To demonstrate this, we train an "inversion model" as described above, i.e., use $\xi_k(x) = \phi_{\text{CLIP}}(x)$ with the same number of trainable parameters and computational budget as for the study in

Figure 9: Text-to-image generalization needs a generative prior or retrieval. See Sec. 4.2.

Fig. 8. When directly using text embeddings for inference, the model renders samples which generally resemble the prompt, but the visual quality is low (CLIP score $0.26 \pm 0.05$, FID $\sim 87$). Modeling the prior with a conditional normalizing flow [18, 62] improves the visual quality and achieves similar results in terms of text-consistency (CLIP score $0.26 \pm 0.3$, FID $\sim 45$), albeit requiring paired data. See Fig. 9 for a qualitative visualization and Appendix F.2.1 for implementation and training details.

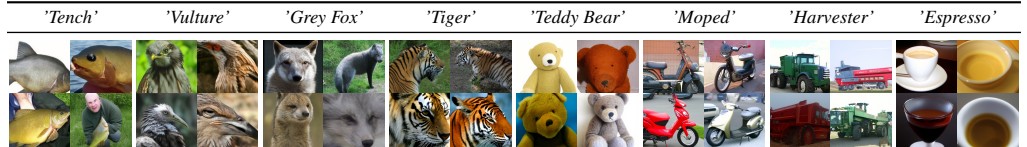

'Tench'   'Vulture'   'Grey Fox'   'Tiger'   'Teddy Bear'   'Moped'   'Harvester'   'Espresso'

Figure 10: *RDM* can be used for class-conditional generation on ImageNet despite being trained without class labels. To achieve this during inference, we compute a pool of nearby visual instances from the database $\mathcal{D}$ for each class label based on its textual description, and combine it with its $k-1$ nearest neighbors as conditioning.

**Class-Conditional Synthesis**   Similarly we can apply our model to zero-shot class-conditional image synthesis as proposed in Sec. 3.3. Fig. 10 shows samples from our model for classes from ImageNet. More samples for all experiments can be found in Sec. G.

### 4.3   Zero-Shot Text-Guided Stylization by Exchanging the Database

In our semi-parametric model, the retrieval database $\mathcal{D}$ is an explicit part of the synthesis model. This allows novel applications, such as replacing this database after training to modify the model and thus its output. In this section we replace $\mathcal{D}_{\text{train}}$ of the ImageNet-*RDM* built from Open-Images with an alternate database $\mathcal{D}_{\text{style}}$, which contains all 138k images of the WikiArt dataset [66]. As in Sec. 4.2 we retrieve neighbors from $\mathcal{D}_{\text{style}}$ via a text prompt and use the text-retrieval strategy *iii)*. Results are shown in Fig. 11 (top row). Our model, though only trained on Ima-

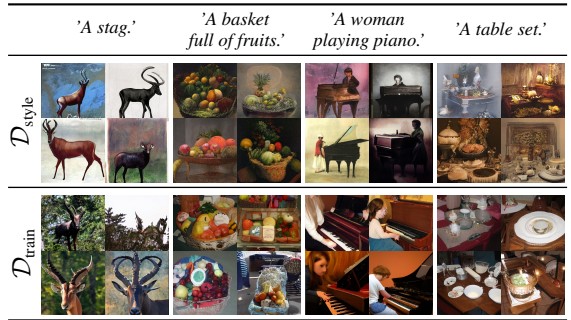

'A stag.'   'A basket full of fruits.'   'A woman playing piano.'   'A table set.'

Figure 11:   Zero-shot text-guided stylization with our ImageNet-*RDM* . Best viewed when zoomed in.

geNet, generalizes to this new database and is capable of generating artwork-like images which depict the content defined by the text prompts. To further emphasize the effects of this post-hoc exchange of $\mathcal{D}$, we show samples obtained with the same procedure but using $\mathcal{D}_{\text{train}}$ (bottom row).

### 4.4   Increasing Dataset Complexity

To investigate their versatility for complex generative tasks, we compare semi-parametric models to their fully-parametric counterparts when systematically increasing the complexity of the training data $p(x)$. For both *RDM* and *RARM*, we train three identical models and corresponding fully parametric baselines (for details *cf.* Sec. F.2) on the dogs-, mammals- and animals-subsets of ImageNet [13], *cf.* Tab. 7, until convergence. Fig. 12 visualizes the results. Even for lower-complexity datasets such as *IN-Dogs*, our semi-parametric models improve over the baselines except for recall, where *RARM* performance slightly worse than a standard AR model. For more complex datasets, the performance gains become more significant. Interestingly, the recall scores of our models *improve* with increasing complexity, while those of the baselines strongly degrade. We attribute this to

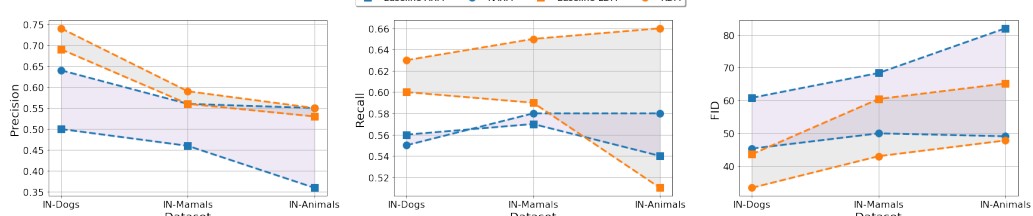

Figure 12: Assessing our approach when increasing dataset complexity as in Sec. 4.4. We observe that performance-gaps between semi- and fully-parametric models increase for more complex datasets.

the explicit access of semi-parametric models to nearby visual instances for *all* classes including underrepresented ones via the $p_\mathcal{D}(\tilde{x})$, *cf.* Eq. (6), whereas a standard generative model might focus only on the modes containing the most often occurring classes (dogs in the case of ImageNet).

## 4.5 Quality-Diversity Trade-Offs

**Top-m sampling.** In this section, we evaluate the effects of the *top-m sampling* strategy introduced in Sec. 3.3. We train a *RDM* on the ImageNet [13] dataset and assess the usual generative performance metrics based on 50k generated samples and the entire training set [5]. Results are shown in Fig. 13a. For precision and recall scores, we observe a truncation behavior similar to other inference-time sampling techniques [5, 15, 32, 23]: For small values of $m$, we obtain coherent samples, which all come from a single or a small number of modes, as indicated by large precision scores. Increasing $m$, on the other hand, boosts diversity at the expense of consistency. For FID and IS, we find a sweet spot for $m = 0.01$, which yields optima for both of these metrics. Visual examples for different values of $m$ are shown in the Fig. 16. Sec. E.5 also contains similar experiments for *RARM* .

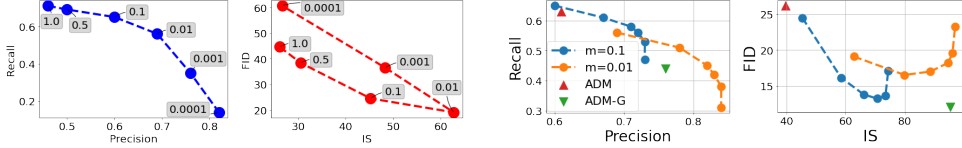

(a) Quality-diversity trade-offs when applying top-m sampling.  (b) Assessing the effects of classifier free guidance.

Figure 13: Analysis of the quality-diversity trade-offs when applying top-m sampling and classifier-free guidance.

**Classifier-free guidance.** Since *RDM* is a conditional diffusion model (conditioned on the neighbor encodings $\phi(y)$), we can apply classifier-free diffusion guidance [32] also for unconditional modeling. Interestingly, we find that we can apply this technique without adding an additional $\emptyset$-label to account for a purely unconditional setting while training $\epsilon_\theta$, as originally proposed in [32] and instead use a vector of zeros to generate an unconditional prediction with $\epsilon_\theta$. Additionally, this technique can be combined with *top-m sampling* to obtain further control during sampling. In Fig. 13b we show the effects of this combination for the ImageNet-model as described in the previous paragraph, with $m \in \{0.01, 0.1\}$ and classifier scale $s \in \{1.0, 1.25, 1.5, 1.75, 2.0, 3.0\}$, from left to right for each line. Moreover we qualitatively show the effects of guidance in Fig. 18, demonstrating the versatility of these sampling strategies during inference.

## 5 Conclusion

This paper questions the prevalent paradigm of current generative image synthesis: rather than compressing large training data in ever-growing generative models, we have proposed to efficiently store an image database and condition a comparably small generative model directly on meaningful samples from the database. To identify informative samples for the synthesis tasks at hand we follow an efficient retrieval-based approach. In the experiments our approach has outperformed the state of the art on various synthesis tasks despite demanding significantly less memory and compute. Moreover, it allows (i) conditional synthesis for tasks for which it has not been explicitly trained, and (ii) post-hoc transfer of a model to new domains by simply replacing the retrieval database. Combined with CLIP's joint feature space, our model achieves strong results on text-image synthesis, despite being trained only on images. In particular, our retrieval-based approach eliminates the need to train an explicit generative prior model in the latent CLIP space by directly covering the neighborhood of a given data point. While we assume that our approach still benefits from scaling, it shows a path to more efficiently trained generative models of images.

## Acknowledgements

This work has been funded by the Deutsche Forschungsgemeinschaft (DFG, German Research Foundation) within project 421703927 and the German Federal Ministry for Economic Affairs and Energy within the project KI-Absicherung - Safe AI for automated driving.

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
