# Appendix

## A Limitations

While our approach boosts performance of both retrieval-augmented AR and diffusion models and significantly lowers the count of trainable parameters compared to their fully-parametric counterparts, our models still have more trainable parameters than other types of generative models, e.g GANs (Tab. 2). Futhermore, we note the long sampling times of both *RDM* and *RARM* compared to single step generative approaches like GANs or VAEs. However, this drawback is inherited from the underlying model class and is not a property of our retrieval-based approach. Neighbor retrieval is fast and incurs negligible computational overhead.

Another limitation is an inherent tradeoff between database size (and associated storage and retrieval costs) and model performance, as evident from Fig. 1. Storing and searching indices for databases of up to billions of images can become quite costly. Furthermore, our approach depends on the image representation chosen to encode images from the retrieval database $\mathcal{D}$ and the retrieval model. Both have significant influence on the performance of the RDM/RARM and further research is needed to determine the best choices here.

Our work demonstrates the benefits of adding an external database in general. However, the choice of the underlying dataset as well as the overall construction strategy of this database is not further investigated. Sec. E.3 analyzes the effect of the patch size, yet these patches are chosen randomly and it is an open question for future research if generating patches from the dataset in a systematic way further improves the obtained results.

Finally, this work does not investigate the scaling behavior of semi-parametric generative modeling. This would be an interesting direction for future work, as we already observe that a model trained only on ImageNet acquires strong zero-shot capabilities, see e.g. Sec. 4.2 and 4.3, although this dataset is small and obtains limited diversity compared other publicly available datasets [71, 73, 46]. Work in NLP [4] suggests that retrieval-augmented transformer models obey a scaling behavior, and we hypothesize that such a property might also exist for image models. The dependence on the CLIP encoder (e.g. ViT-B/32 vs ViT-L/14) should also be investigated in future work.

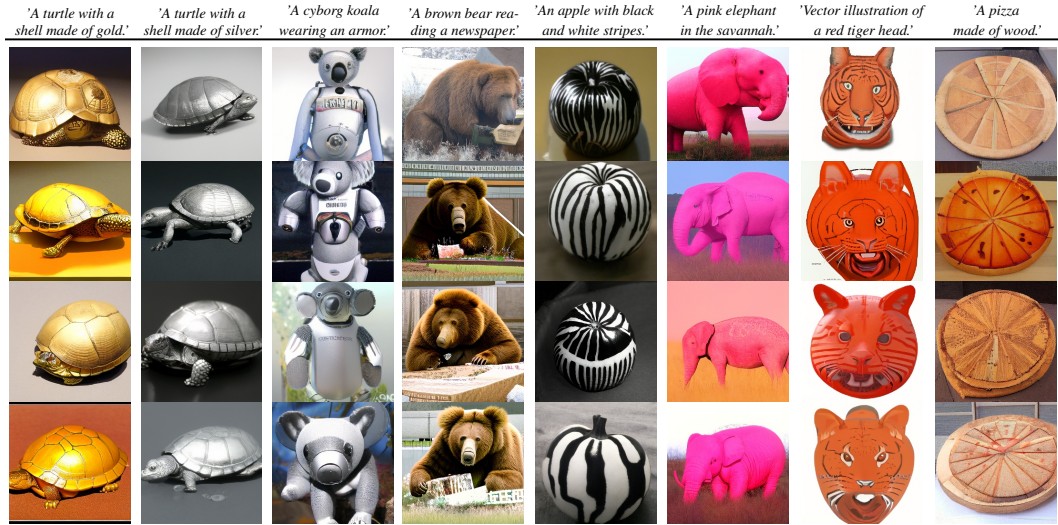

| 'A turtle with a shell made of gold.' | 'A turtle with a shell made of silver.' | 'A cyborg koala wearing an armor.' | 'A brown bear reading a newspaper.' | 'An apple with black and white stripes.' | 'A pink elephant in the savannah.' | 'Vector illustration of a red tiger head.' | 'A pizza made of wood.' |

Figure 14: Additional zero-shot text to image samples from our model as in Fig. 2. Samples are generated with classifier-free scale $s = 2.5$ and 100 DDIM steps.

## B Societal Impacts

Large-scale generative image models enable creative applications and autonomous media creation, but can also be viewed as a dual-use technology [14] with negative implications. A notorious example

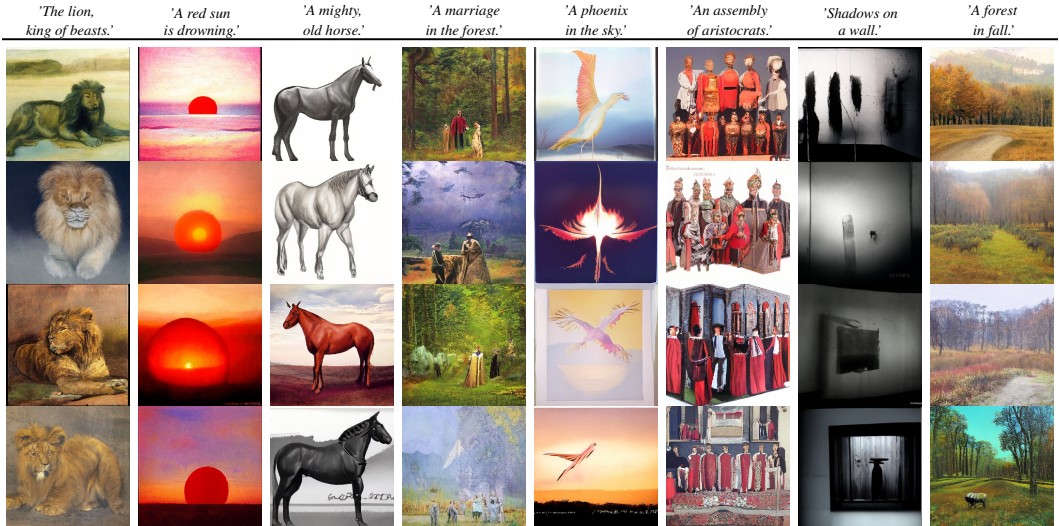

| 'The lion, king of beasts.' | 'A red sun is drowning.' | 'A mighty, old horse.' | 'A marriage in the forest.' | 'A phoenix in the sky.' | 'An assembly of aristocrats.' | 'Shadows on a wall.' | 'A forest in fall.' |

Figure 15: Additional samples for zero-shot text-guided stylization with our ImageNet *RDM* as in Fig. 11. Samples are generated with classifier-free scale $s = 2.5$ and 100 DDIM steps.

are so-called "deep fakes" that have been used, for example, to create pornographic "undressing" applications [14]. Furthermore, the immediate availability of mass-produced high-quality images can be used to spread misinformation and spam, which in turn can be used for targeted manipulation in social media [14, 24].

Datasets are crucial for deep learning as they are the main input of information. For our model, this concerns the data used in training and inference, as the retrieval database can be considered as a part of the model. Therefore, the diversity and bias of the synthesized images depends heavily on the diversity and bias in these datasets. For example, a bias of representing a particular skin tone or gender imbalance (i.e., a lack of diversity) already present in the datasets can be easily amplified by deep learning models trained on it [20, 36, 82]; and the effect of post-training truncation models on these phenomena remains under-explored. However, we note that quantitative diversity analysis of our retrieval-based approach shows that it better covers the data distribution, resulting in less bias towards certain modes in the datasets, such as overrepresented communities, and might be a step towards more balanced and controllable generative models.

Furthermore, one should consider the ability to curate the database to exclude (or explicitly contain) potential harmful source images. When creating a public API that approach could offer a cheaper way to offer a safe model than retraining a model on a filtered subset of the training data or doing difficult prompt engineering. Conversely, including only harmful content is an easy way to build a toxic model.

Large-scale image datasets that are used to train advanced synthesis models are usually scraped from the internet [71, 72], and the ethical implications of training on, for example, original digital artwork remain an open question. In addition, it is difficult to assess what impact a single training image had on a generated image or the final generative model.

That is in contrast to the image database used for the retrieval algorithm: Here, retrieved images have a discernible effect on the output, and the database used during inference may only consist of relatively few high quality images. Therefore, this could allow for attribution and compensation of the involved content creators. As an example, when providing an online interface for a retrieval augmented synthesis model, that cost can be factored in together with the hardware costs and be automatically paid for each generated image. However, the extent to which retrieved representations alone contribute to the final model output needs further investigation.

Lastly, training large image synthesis models with millions of parameters using specialized hardware[4] requires significant financial investment and is therefore available only to a limited number of

---

[4]See section: F.2 for details on the hardware used for the experiments in this work

institutions. The limited access to these large models becomes particularly problematic if these powerful models are not made freely available[5] after training and remain exclusively in the hands of these same institutions, hindering full exploration of their capabilities and biases.

## C Concurrent Work

Very recently, two concurrent approaches related to our work, unCLIP [59] and kNN-Diffusion [2], have been proposed. unCLIP produces high quality text-image results by conditioning a diffusion model on the image representation of CLIP [57] and employing large-scale computation. However, unlike our work, it conditions on the CLIP representation of the training image itself, which makes it necessary to learn a generative text-image prior in the CLIP space later. We show that the neighbor-based approach provides an alternative to training a generative prior to translate between CLIP embeddings (see Sec. 4.2 and Fig. 14). Our approach allows to modify the retrieval database $\mathcal{D}$ *after* training, which can be used to control the style of the rendered samples (Sec. 4.3, Fig. 15). We also show that unCLIP can be interpreted as a special case of our formulation *without* an external database and the retrieval strategy $\xi_k(x) = \{\phi_{\text{CLIP}}(x)\}$, for which we train a conditional normalizing flow as the generative prior (see Sec. 4.2).

kNN-Diffusion, like our approach, avoids this problem by conditioning on a neighborhood of the image. Although both kNN-Diffusion and our approach are fundamentally very similar, we use a continuous rather than a discrete diffusion formulation, analyze different forms of neighborhood representations, investigate autoregressive models in addition to diffusion models and are not exclusively limited to text-image synthesis.

## D Trading Quality for Diversity

Here, we present additional details on top-m sampling and further elaborate on the classifier-free guidance technique for *RARM* .

### D.1 Further Details on Top-m Sampling

Many approaches to (conditional) generative modeling offer ways to trade off sample quality for diversity at test time. GANs and diffusion models can achieve this by leveraging conditional information via *truncated sampling* [5] and *classifier guidance* [15, 32], while models based on a categorical distribution like most autoregressive models allow for *top-k sampling* [23].

We propose a similar technique for semi-parametric generative models. Let $Z_m = \sum_{\tilde{x} \in \mathcal{D}^{(m)}} p_{\mathcal{D}}(\tilde{x})$, where $\mathcal{D}^{(m)} \subseteq \mathcal{D}$ is the subset containing the fraction $m \in (0, 1]$ of most likely examples $\tilde{x} \sim p_{\mathcal{D}}(\tilde{x})$. Similar to top-k sampling, we define a truncated distribution

$$\mu(\tilde{x}) = \begin{cases} p_{\mathcal{D}}(\tilde{x})/Z_m \,, & \text{if } \tilde{x} \in \mathcal{D}^{(m)} \\ 0 \,, & \text{else} \,, \end{cases} \tag{7}$$

which we can use as proposal distribution to obtain $\mathcal{P}$ according to Eq. (6). Thus, for small values of $m$, this yields samples from a narrow, almost unimodal distribution. Increasing $m$ on the other hand, increases diversity, potentially at the cost of reduced sample quality. We analyze this trade-off in Sec. 4.5 and show corresponding visual samples in Fig. 16 and Fig. 17. In analogy to top-k sampling, we dub this sampling scheme *top-m sampling*.

To gain additional flexibility during inference, this scheme can further be combined with model-specific sampling techniques such as **classifier-free diffusion guidance** [32], since our model *RDM* is a conditional diffusion model of of the nearest neighbor encodings $\phi(y)$. We present results using different combinations of $m$ and classifier-free guidance scales $s$ in Sec. 4.5. Moreover, we show accompanying visual examples for the effects of classifier-free unconditional guidance in Fig. 18.

---

[5]Often a publication of the trained model weights or of the source code is rejected with reference to the dual-use properties listed above.

| Single Example | $m = 10^{-5}$ | $m = 10^{-4}$ | $m = 10^{-3}$ | $m = 10^{-2}$ | $m = 10^{-1}$ |
|---|---|---|---|---|---|

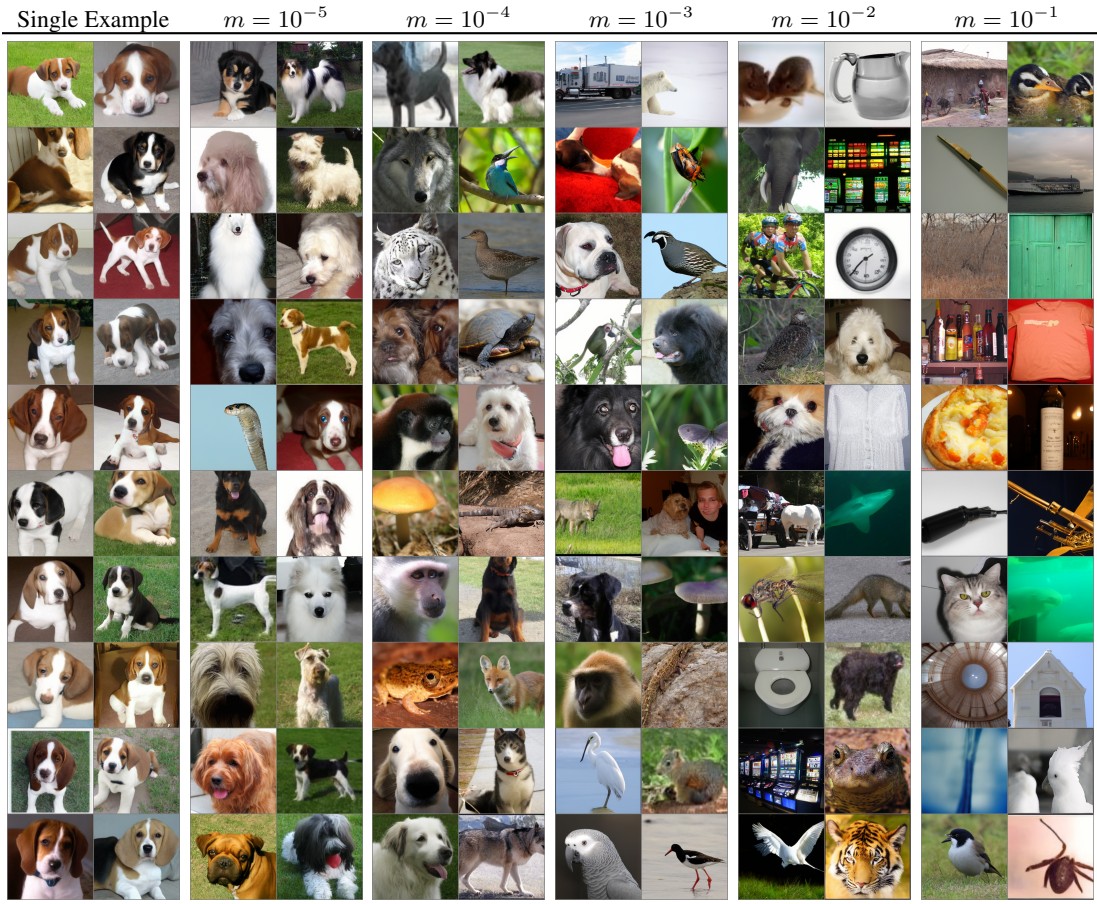

Figure 16: Visual examples on the quality-diversity trade off obtained by *top-m sampling*. For heavily truncated $p_{\mathcal{D}}(\tilde{x})$ we obtain extremely low sample diversity as visualized in the examples on the left part. Increasing $m$ results in more diversity but lower sample fidelity (right part). All images generated with guidance scale $s = 1.5$ and 100 DDIM steps.

## D.2 Classifier-free Guidance for *RARM*

Classifier-free guidance [32] was originally proposed for conditional diffusion models, nonetheless, it can also be applied to conditional autoregressive transformers [12]. We find that, similar to the diffusion head, (*cf.* Sec. 4.5) it is sufficient to condition the *RARM* on a zero representation to gain an improvement using classifier-free guidance during test time without additional unconditional training. Given previous image tokens $t_1, \ldots, t_{k-1}$ guidance can then be applied as

$$
\begin{aligned}
&\log(p_{\text{cfg}}(t_k \mid t_{<k}, \{y_i\}_i)) \\
&= \log(p_\theta(t_k \mid t_{<k}, \{0\})) + s \cdot \Big( \log(p_\theta(t_k \mid t_{<k}, \{y_i\}_i)) - \log(p_\theta(t_k \mid t_{<k}, \{0\})) \Big).
\end{aligned} \tag{8}
$$

Qualitative samples obtained with this strategy are depicted in Fig. 19.

| Single Example | $m = 10^{-5}$ | $m = 10^{-4}$ | $m = 10^{-3}$ | $m = 10^{-2}$ | $m = 10^{-1}$ |
|---|---|---|---|---|---|

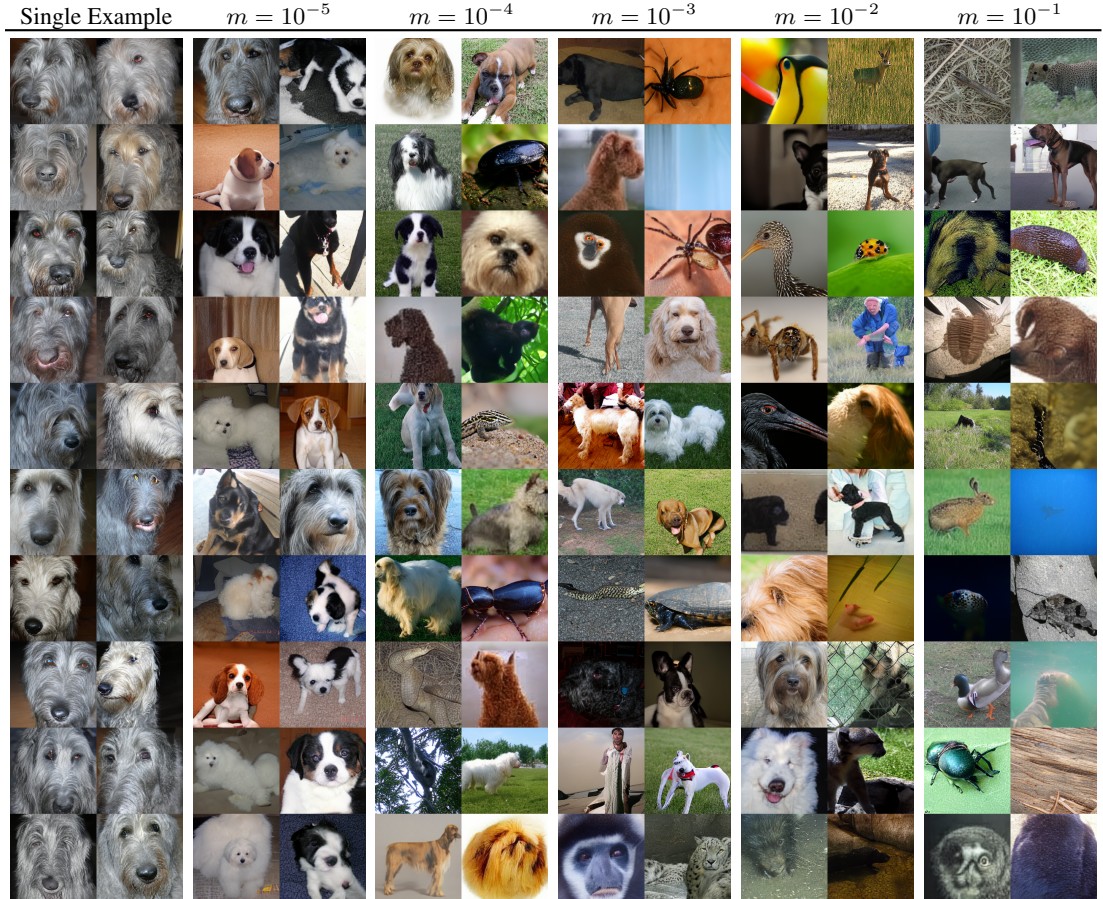

Figure 17: Visual examples on the quality-diversity trade off obtained by *top-m sampling* using our *RARM* trained on IN-animals. For heavily truncated $p_{\mathcal{D}}(\tilde{x})$ we obtain extremely low sample diversity as visualized in the examples on the left part. Increasing $m$ results in more diversity but lower sample fidelity (right part). All images generated with guidance scale $s = 2.0$ and generated with top-$k = 4096$. Note that this model is trained on the Animals subset of ImageNet. Therefore, the proposal distribution $p_{\mathcal{D}}(\tilde{x})$ differs from that of the shown results for *RDM* in Fig. 16, which is trained on the entire ImageNet dataset. This is the reason for the different classes of dogs for the leftmost column of this Figure compared to Fig. 16.

# E  Additional Experiments

## E.1  Detailed Evaluation on Zero-Shot Stylization

### E.1.1  Quantitative Evaluation

In this section we quantitatively evaluate the zero-shot stylization capabilities of *RDMs* presented in Sec. 4.3 and explore their limitations on that task. First, we assess the post-hoc steerability of *RDMs* by exchanging the database at inference time and compare it with that of IC-GAN. We use WikiArt [66] as inference database both for our ImageNet *RDM-OI* and for the publicly released IC-GAN trained on ImageNet[6] and generate 50K examples with each model.

| Method | FID↓ | | Precision↑ | | Recall↑ | |
|---|---|---|---|---|---|---|
| Backbone | *I-V3* | *CLIP* | *I-V3* | *CLIP* | *I-V3* | *CLIP* |
| IC-GAN | 24.75 | 35.17 | 0.47 | 0.38 | 0.28 | 0.02 |
| *RDM-OI* | **21.50** | **13.01** | **0.63** | **0.46** | **0.34** | **0.11** |

Table 4: Performance metrics evaluated against examples from WikiArt for IC-GAN and *RDM-OI* trained on ImageNet. During inference both models are conditioned on samples from the WikiArt database.

By computing FID, Precision and Recall scores against WikiArt, we can measure how well the two models approximate the WikiArt image manifold. From Tab. 4 we can see that *RDM* outperforms IC-GAN on all metrics.

---

[6]Code and model taken from `https://github.com/facebookresearch/ic_gan`

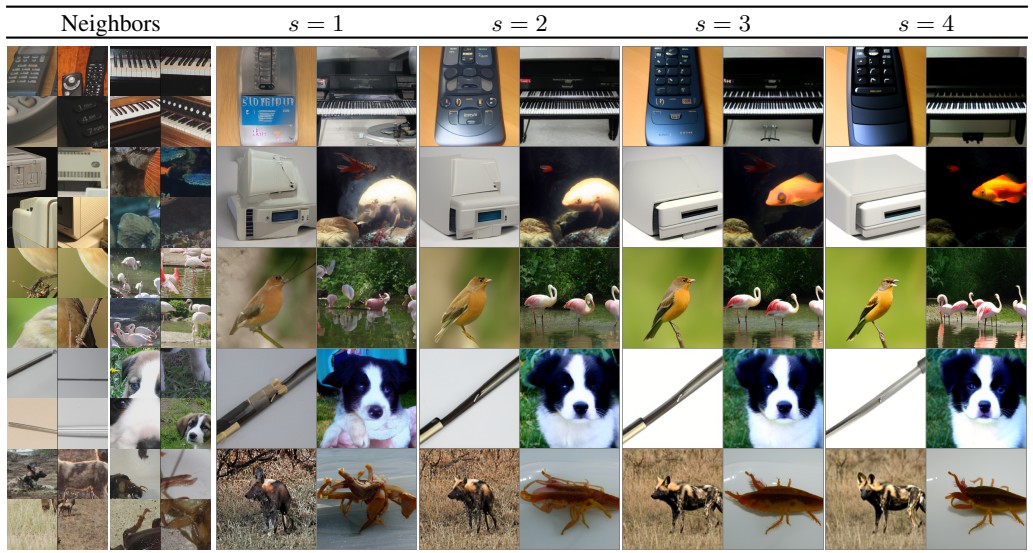

Figure 18: Visualizing the effects of retrieval based classifier free guidance. All images generated with fixed random seed, $m = 0.1$ and 100 DDIM steps.

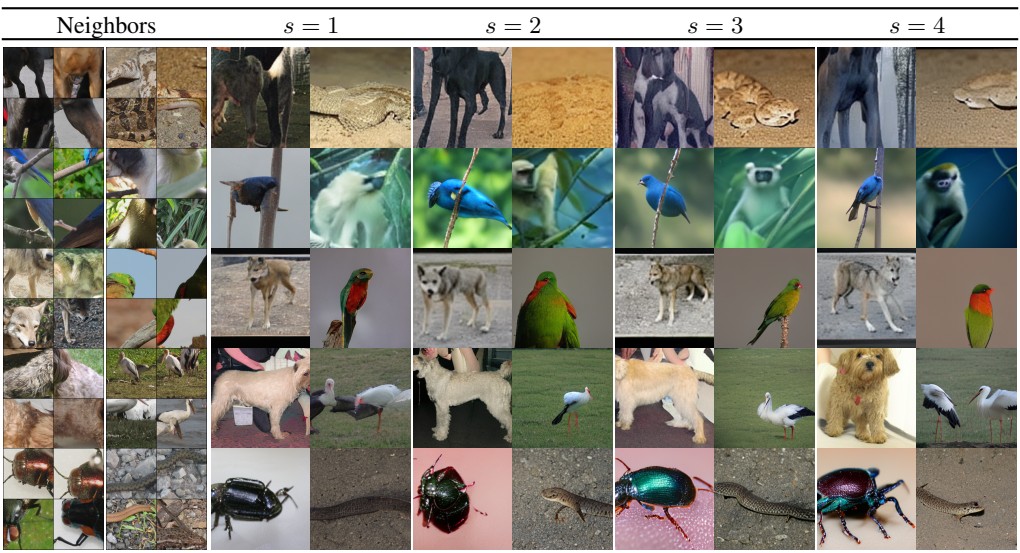

Figure 19: Visualizing the effects of retrieval based classifier-free guidance for the *RARM* trained on IN-animals. All images generated with fixed random seed, $m = 0.01$ and top-$k = 4096$.

| | *RDM* | | | IC-GAN [8] | |
| $\mathcal{D}_{\text{train}}$ | Pacs Cartoon | WikiArt | $\mathcal{D}_{\text{train}}$ | Pacs Cartoon | WikiArt |

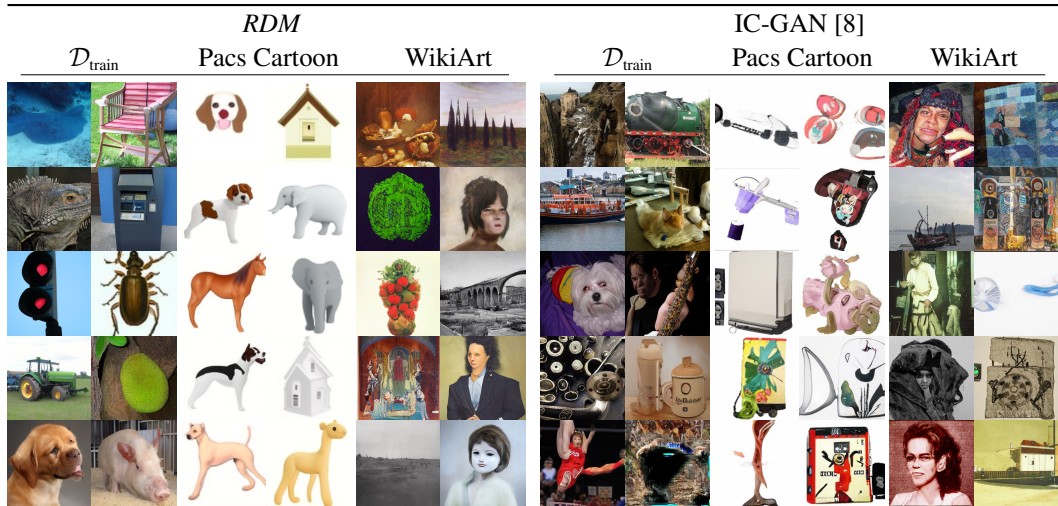

Figure 21: Direct comparison of samples from *RDM* with those of IC-GAN on i) the train-time database $\mathcal{D}_{\text{train}}$ which is the training set of ImageNet for IC-GAN and ii) on the

Since it better approximates the target image manifold, we conclude that our model can better adopt the properties of this novel database during inference.

Furthermore to explicitly compare how well the two models preserve properties of the inference-time database, we train a linear-probe on ResNet-50 features to distinguish between images from these two datasets. The resulting classifier achieves an accuracy of 96% on an unseen validation set. We measure its accuracy on the 50K generated images for each class from both methods and show the obtained results in Fig. 20. We see that for both ImageNet and WikiArt, a higher percentage of images generated by *RDM* are classified as belonging to the respective dataset, thus showing *RDMs* to better adopt those databases' properties.

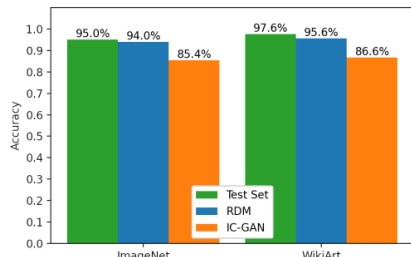

Figure 20: Evaluating accuracy of a binary classifier trained distinguishing between WikiArt and ImageNet on generated samples for IC-GAN and *RDM-OI* .

### E.1.2 Qualitative Evaluation

We also show a qualitative side-by-side comparison between *RDM* and IC-GAN in Fig. 21 when using the respective train databases (left), the PacsCartoon dataset [49] (mid) and WikiArt (right) as inference database. It shows that *RDM* not only achieve higher visual quality on the task it was trained for, i.e. generation on ImageNet, but also that the generated images based on the novel, exchanged inference contain significantly more properties of the respective databases than those of IC-GAN. However, we also see that *RDMs* struggle to generate realistic examples when conditioned *semantic* concepts they have never seen in the training as visible for the 'giraffe' cartoon sample in the fourth column of the bottom row.

### E.2 Alternative Image Encoders $\phi$

As conditioning on raw image pixels would result in excessive memory/storage demands, finding an appropriate compressed representation $\phi(y)$ for the retrieved neighbors $y \in \mathcal{M}_{\mathcal{D}}^{(k)}$ is of central importance. For our main experiments we implement $\phi$ with the CLIP image encoder as its embedding space is compact and shared with the text-embeddings of the CLIP text encoder. There are principally many other choices of $\phi$ possible, including learning it jointly together with the decoding head. However, since representations pretrained on a large corpus of data has proven not only train-time memory efficient but also beneficial for image generation, we here focus on such pretrained feature extractors.

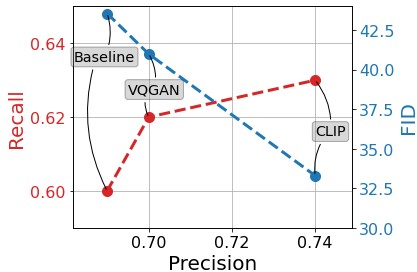

Figure 22: Performance of *RDM* with different nearest neighbor representations.

We investigate two types of representations and compare those from a pre-trained VQGAN encoder [22], representations from the image encoder of CLIP [57]. For these experiments we focus on *RDM* , i.e. we implement the decoding heads of the compared models as a conditional diffusion model. For both compared models we use $k = 4$ nearest neighbors during training and inference. Moreover we compare them with a full-parametric *LDM* baseline with $1.3\times$ more parameters. To render training less compute intensive, we train them on the ImageNet-dogs subset (see Sec. F.2.3).

Fig. 22 summarizes the obtained results which again demonstrate the efficacy of semi-parametric generative modeling compared to fully-parametric models, as both VQGAN-[7] and CLIP-nearest-neighbor encodings improve sample quality (higher precision [47], lower FID [31]) as well as diversity (higher recall [47]), despite using less trainable parameters (the baseline uses $1.3\times$ more parameters). Moreover we see that the model conditioned on CLIP image embeddings consistently improves over that which uses VQGAN encodings. Thus we use such models for our experiments in the main paper.

### E.3   Patch Size of Images in the Database

Our retrieval database consists of 20M examples originating from the OpenImages [46], see Sec. F.1 for details. As the images in OpenImages are much larger than our train time image size of 256 pixels per side, we crop multiple patches per image. For the train database used for the models presented in the main experiments we use a patch size of $256 \times 256$ pixels. However, since the chosen patch size determines the properties of the images in the database[8] we investigate the effects of varying the size of the extracted patches in the database.

To this end we train three identical *RDM* with $k = 4$ with databases consisting of patches which were extracted from OpenImages by using different patch size $H_{\mathcal{D}} = W_{\mathcal{D}} \in \{64, 128, 256\}$. As in Sec. E.2 we train the models on the dogs-subset of ImageNet compare the semi-parametric models with an *LDM* baseline with $1.3\times$ more trainable parameters.

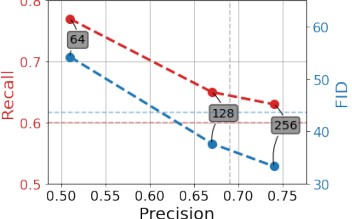

Fig. 23 visualizes the obtained results. Vertical and horizontal bars denote the performance of the *LDM* baseline. As expected, we observe the patch size to have substantial influence on the performance of semi-parametric models. We see that an patch

Figure 23: Effect of patch size of images in the retrieval database.

size of 64 pixel seems to be too small, resulting in worse performance compared to the baseline. Increasing the patch size results in significant improvements over the baseline, despite a smaller parameter count. High precision [47] and FID [31] indicate that conditioning on larger patches results in improved sample quality. Recall values [47] decrease when increasing the patch size. This is due to the fact that for the model with a patch size of 64, the generated samples lack perceptual consistency, as indicated by the small precision values. However the model with a database of patch size 256 still has a recall score $> 0.60$ which is still high and clearly larger than the achieved value of the baseline. This demonstrates that retrieval-augmented models maintain high sample diversity and conditioning on global object attributes yields more coherent samples than only using local object parts in the database. In the future, increasing the patch size beyond 256 px per side bears potential to further improve sample quality achieved by semi-parametric models.

---

[7]For more details on how we feed this representation to the decoding head via cross attention, see Sec. F.3.5.

[8]Larger patch sizes will result in more images depicting objects as a whole, whereas smaller patch sizes will rather show object parts.

### E.4 Optimizing $k_{\text{train}}$ for *RARM*

Similar to Sec. 4.1 we here evaluate suitable choices of $k_{\text{train}}$ for *RARM* and therefore train models with the same decoding head but with different $k_{\text{train}} \in \{1, 2, 4, 8, 16\}$ on the ImageNet dogs subset. We show the resulting evaluation metrics computed based on 2000 samples in Fig. 24, where we observe the models with $k_{\text{train}} \in \{2, 4\}$ to perform best as both models yield good FID scores while sill achieving comparably high precision and recall values. The optimal choice seems to be $k_{\text{train}} = 2$ which is different than for *RDM*, where we found $k_{\text{train}} = 8$ to yield the best results.

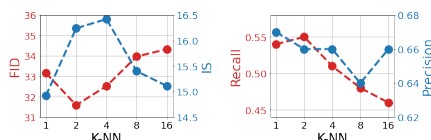

Figure 24: Effect of $k_{\text{train}}$ for *RARM*.

### E.5 Top-m Sampling for *RARM*

In this section we analyze the effects of top-m sampling for *RARM* similar to the evaluation for *RDM* presented in Fig. 13a. To this end we use the best performing model for $k_{\text{train}} = 2$ from Sec. E.4 and generate 10000 samples for $m \in \{1., 0.5, 0.1, 0.01, 0.001, 0.0001\}$ without classifier-free guidance. Fig. 25 visualizes the results which show the same truncation behavior as observed for *RDM*, see Fig. 13a, including the FID-IS sweet spot

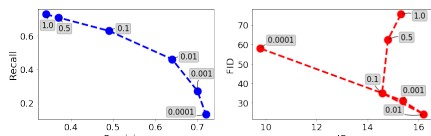

Figure 25: Quality-diversity trade-offs when applying top-m sampling with *RARM*.

at $m = 0.01$. This experiment provides further evidence for the discussed advantages of semi-parametric generative models compared to their fully-parametric counterparts, irrespective of the actual realization of the decoding head.

## F Implementation Details

### F.1 Train-time Database and Retrieval Strategy

As mentioned in the main paper, we build our database from the OpenImages dataset [46], which contains 9M images of varying spatial sizes with a shorter edge length of at least 1200 px. To build our 20 M images database we resize all images such that the shorter edge length is equal to 1200 px and subsequently randomly select 2-3 patches of size $256 \times 256$ px per image of OpenImages. Thus, we use each of these images at least once. We investigate the effects of using different patch sizes for building the database in Sec. E.3.

For all datasets investigated in the work, we precompute $k = 20$ neighors for each query image of a given train dataset and store the resulting CLIP-embeddings along with the image ids and patch coordinates of the corresponding image in the OpenImages dataset. This allows us to also visualize the images corresponding to the neighbors in the CLIP space.

For nearest neighbor retrieval we use the ScaNN search library [28]. With this choice, retrieving 20 nearest neighbors from the database described above takes approximately 0.95 ms. Thus, including NN retrieval in the training process would also not mean significant training time overheads.

### F.2 Training Details

#### F.2.1 Models with Diffusion Based Decoding Heads

In Tab. 5 we show the hyperparameters which were used to train our presented models, which use diffusion based decoding heads. For the retrieval-augmented models, the hyperparameters correspond only to the decoding head, as the other parts of the model are not trainable. We trained our main model (which was used to generate all qualitative results in this work as well as the quantitative results shown in Tab. 2) on eight NVIDIA A-100-SXM4 with 80GB RAM per GPU. The overall training time compute spent to train this model is 48 A-100 days when considering a single A-100 with 80 GB RAM or 96 A-100 days when calculating with an A-100 with 40GB.

The models evaluated in the $k_{\text{train}}$ experiments in Sec. 4.1 and Sec. 4.2 are all trained on two NVIDIA A-100-SXM4 with 80GB RAM per GPU for the same number of train steps. To enable larger batch

size we only parameterize these models with 200M trainable parameters and use a compression model which is trained with KL-redularization with a downsampling factor $f = 16$. For a detailed explanation of the compression models and of the *LDM* framework, see [63]. This is in contrast to our other diffusion based models, which use a VQ-regularized compression model with $f = 4$, as $f = 16$ allows us to further increase the batch size and thus result in faster converging models. The normalizing flow used to model the CLIP generative prior in Sec. 4.2 uses a "modernized" version of the invertible backbone, built from 200 blocks that consist of coupling layers [17], activation normalization [45] and shuffling as in [62, 3]. We replace batch normalization in the sub-networks with layer normalization and RELU with GELU [30] nonlinearities.

The models from the analysis using the subsets of ImageNet in Sec. 4.4 are all trained on a single NVIDIA A-100 GPU with 40GB RAM. To be able to use the same batch sizes also for the *LDM* baselines shown in these experiments, each of which has 1.3 times more parameters than the corresponding *RDM*, we use gradient checkpointing [11] to reduce memory cost during backpropagation at the expense of additional computations in the forward pass. As these baselines are 'common' unconditional models, we use self-attention (SA) instead of the cross-attention layers (CA) which are used to feed the nearest neighbor representation $\phi$ to the decoding head of the semi-parametric models. All our models are implemented in PyTorch. We will release the code and pretrained models in the near future.

| | *RDM*[*] | *RDM*[†] | *RDM*[‡] | baseline *LDM*[‡] |
|---|---|---|---|---|
| Dataset | ImageNet (IN) | ImageNet | IN-subsets, *cf.* Tab. 7 | IN-subsets, *cf.* Tab. 7 |
| $z$-shape | $64 \times 64 \times 3$ | $16 \times 16 \times 16$ | $64 \times 64 \times 3$ | $64 \times 64 \times 3$ |
| $|\mathcal{Z}|$ | 8192 | KL | 8192 | 8192 |
| Diffusion steps | 1000 | 1000 | 1000 | 1000 |
| Noise Schedule | linear | linear | linear | linear |
| Model Size | 400M | 200M | 400M | 576M |
| Channels | 192 | 192 | 192 | 224 |
| Depth | 2 | 2 | 2 | 2 |
| Channel Multiplier | 1,2,3,5 | 1,2,2,4 | 1,2,3,5 | 1,2,4,6 |
| BigGAN [5] up/downsampling | ✗ | ✗ | ✗ | ✓ |
| activation rescaling [79, 38, 39] | ✗ | ✗ | ✗ | ✓ |
| Number of Heads | 32 | 32 | 32 | 32 |
| Batch Size | 1240 | 640 | 56 | 56 |
| Iterations | 112K | 240K | subset dependent[§] | subset dependent[§] |
| Learning Rate | 1.0e-4 | 1.0e-4 | 1.0e-4 | 1.0e-4 |
| Conditioning | CA | CA | CA | - |
| CA/SA-resolutions | 32, 16, 8 | 16, 8, 4 | 32, 16, 8 | 32, 16, 8 |
| Embedding Dimension | 512 | 512 | 512 ($\phi = \phi_{\text{CLIP}}$)/1024 ($\phi = \phi_{\text{VQGAN}}$) | - |
| Transformers Depth | 1 | 1 | 1 | - |

Table 5: Hyperparameters for the diffusion based models presented in this work.[*]: All qualitative examples in this work and the numbers presented in Tab. 2 are generated with this model;[†]: The models trained for the $k_{\text{train}}$ experiments in Sec. 4.1 are all trained with these hyperparameters;[‡]: The various semi- and fully-parametric models referred to in Sec. 4.4 are trained with these hyperparameters; [§]: All models were trained until convergence.

### F.2.2 Models with Autoregression Based Decoding Heads

In Tab. 6 we show the hyperparameters which were used to train the autoregressive models presented in this work. For the retrieval-augmented models, the hyperparameters correspond only to the decoding head, as the other parts of the model are not trainable. All autoregressive models are decoder-only GPT-like transformer models and use the same VQGAN compression model with a downsampling factor of $f = 16$. Using such a compression model and applying raster scan reordering [86] results in an input sequence of length 256 for an image of spatial size $256 \times 256$. This prevents our models from allocating excessive amounts of GPU memory, what can arise for long sequences, due to the quadratic complexity of the attention mechanism. The *RARM* have an additional cross-attention block (CA) behind every self-attention block that is used to feed the nearest neighbor representation $\phi$ to the decoding head. We train all autoregressive models on a single NVIDIA A-100 with 40GB RAM.

|  | *RARM* | baseline ARM |
|---|---|---|
| Dataset | ImageNet-Subsets | ImageNet-Subsets |
| Image size | $256 \times 256 \times 3$ | $256 \times 256 \times 3$ |
| Z-shape | $16 \times 16 \times 256$ | $16 \times 16 \times 256$ |
| #Codes | 16 384 | 16 384 |
| Model Size | 231 M | 265 M |
| #Heads | 12 | 14 |
| Channel per Head | 64 | 64 |
| Depth | 18 | 18 |
| Batch Size | 100 | 100 |
| Iterations | subset dependent | subset dependent |
| Learning rate | 5.0e−4 | 5.0e−4 |
| Conditioning | CA | - |
| Context Dimension | 512 | - |

Table 6: Hyperparameters for the autoregressive models used in this work. Qualitative examples and quantitative results stem from different models as described in the corresponding section. All models were trained until convergence.

### F.2.3 Statistics for ImageNet subsets

In Tab. 7 we present detailed statistics for the datasets involved in the comparison of fully- and semi-parametric generative models for increasing complexity of the modeled data distribution. For the dogs subset, we used the class labels ranging from 181 to 280, resulting in a training dataset containing $N = 163K$ examples. Including all mammals lead to overall 241 classes with $N = 309K$ examples whereas training on all 398 classes referring to animals resulted in a dataset of $N = 511K$ individual images. As for our main experiments, we did not use any class labels for training the models on these datasets.

| Dataset | class labels | $N$ |
|---|---|---|
| IN-dogs | 151-280 | 163K |
| IN-mammals | 147-388 | 309K |
| IN-animals | 0-397 | 511K |

Table 7: Statistics for the ImageNet subsets used in the analysis on dataset complexity in Sec. 4.4 and Fig. 12.

### F.3 Evaluation Details

### F.3.1 Analysis Experiments on Effects of $k_{\text{train}}$ from Sec. 4.1

To generate the results shown in the $k_{\text{train}}$ analysis presented Sec. 4.1 we used $m = 0.01$ and no guidance for all compared choices of $k_{\text{train}}$. we assessed performance metrics based on 1000 samples for each run.

### F.3.2 Comparison with State of the Art

For the SOTA comparison presented in Sec. 4.1, we use the evaluation protocol proposed in ADM [15], where performance metrics are calculated based on 50K samples and by using the ImageNet train set as a reference for the data distribution. We also use their publicly available evaluation implementation to obtain comparable results[9]. To be able to compare our models also with IC-GAN [8], which uses train set instances during evaluation, we additionally follow their protocol of evaluating against the validation split. Moreover, we compute precision and recall scores for their method, by using the publicly available pretrained weights[10] for both train and validation splits, see Tab. 2. The low recall scores indicate their generated samples to lack diversity and their GAN based model to only capture few modes of the data distribution, which is a well-known issue for GANs [80, 1, 55, 50]. In contrast, since our models profit from the mode-covering property of the likelihood based objective, our recall scores are sufficiently high for all presented combinations of sampling parameters.

---

[9]https://github.com/openai/guided-diffusion
[10]https://github.com/facebookresearch/ic_gan

### F.3.3 Details on Evaluations on Text-to-Image Generalization

In Sec. 4.2 we evaluate the the generalization capabilities of our ImageNet *RDM* , which is trained only on images, when applied to text-to-image synthesis. For generating the ImageNet-FIDs presented in Fig. 8 we used 2000 samples generated with top-m $= 0.01$ and without unconditional classifier-free guidance. The presented scores for text-to-image synthesis on COCO were synthesized with top-m $= 0.01$ and classifier-free guidance scale $s = 2.0$ for all models. We furthermore applied the same sampling parameters when generating results with the model directly conditioned on CLIP representation, which includes a flow prior for closing the mismatch between CLIP text- and image-embeddings.

### F.3.4 Details on Experiments regarding Dataset Complexity

For both *RDM* and *RARM* we compute the metrics presented in Fig. 12 based on 1000 samples for each individual dataset and use $k = 4$. We also compute the metrics for the fully-parametric baselines with 1000 samples. For *RDM* , we use top-m $= 0.01$ and no classifier-free guidance. For *RARM* we use top-m $= 0.005$, top-$k = 2048$ and no classifier-free guidance.

### F.3.5 Building a Conditioning Sequence with VQGAN-encodings

In the comparison regarding different encoders $\phi$ in Sec. E.2 we compare CLIP image embeddings with those extracted by a pretrained VQGAN encoder. However, for the latter, which yields a three-dimesional tensor for each retrieved nearest neighbor, we have to apply a reshaping to obtain a sequence, which is suitable for being fed to the decoding head via cross-attention. We here implement $\phi$ with a $f16$ VQGAN-encoder pretrained on OpenImages [63][11]. For the default VQGAN input size, which is 256, the latent code of each retrieved neighbor would be of size $16 \times 16 \times 256$. Thus, to further shrink to dimensionality of this representation we resize the input images for each of the $k = 4$ nearest neighbors to $128 \times 128$ px, since this does not hurt the model's performance, resulting in a latent tensor of shape $8 \times 8 \times 256$. We then form a sequence shape $64 \times 256$ for each nearest neighbor representation by applying raster scan reordering [86] and subsequently concatenate all $k = 4$ individual representation channel-wise, resulting in the final conditioning input for the decoding head with a shape of $64 \times 1024$ which can be fed via cross attention.

## G  Additional Samples

In this section we show additional qualitative samples for all presented experiments in the main paper. Fig. 14 shows additional samples of the generalization of our ImageNet *RDM* , when using CLIP-representations of text prompts as inputs, as in Fig 2. Fig. 15 shows additional examples of text-guided stylization with by changing the database for the model ImageNet model mentioned above. With this zero-shot stylization model, we can also generate unconditional samples. This is visualized in Fig. 26 and compared with unconditional samples from the same model, with the original database $\mathcal{D}_{\text{train}}$, which is used during training. We furthermore show additional unconditional samples in Fig. 27 and also more class-conditional samples similar to Fig. 10 in Fig. 29. Additional samples from our experiment which compares the direct use of CLIP text-embeddings and embeddings from a conditional normalizing flow (as in Sec. 4.2) are depicted in Fig. 30. Random samples from the autoregressive models are shown in Fig. 28.

---

[11]`https://github.com/CompVis/latent-diffusion`

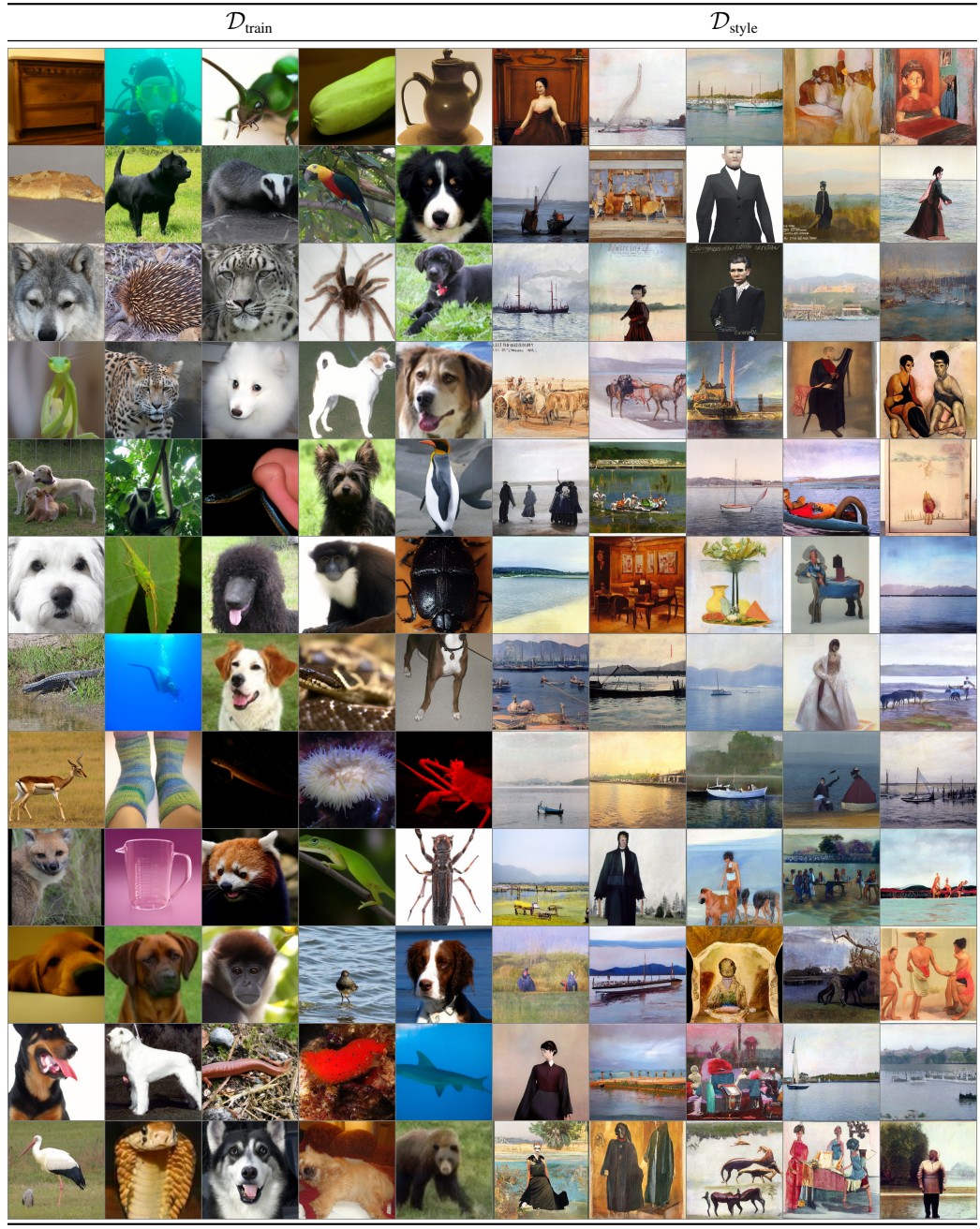

Figure 26: Comparing random unconditional samples when replacing the train database $\mathcal{D}_{\text{train}}$ with a new database $\mathcal{D}_{\text{style}}$ consisting of the entire image corpus of WikiArt [66]. Images were generated with classifier-free scale $s = 2.0$ and 100 DDIM steps.

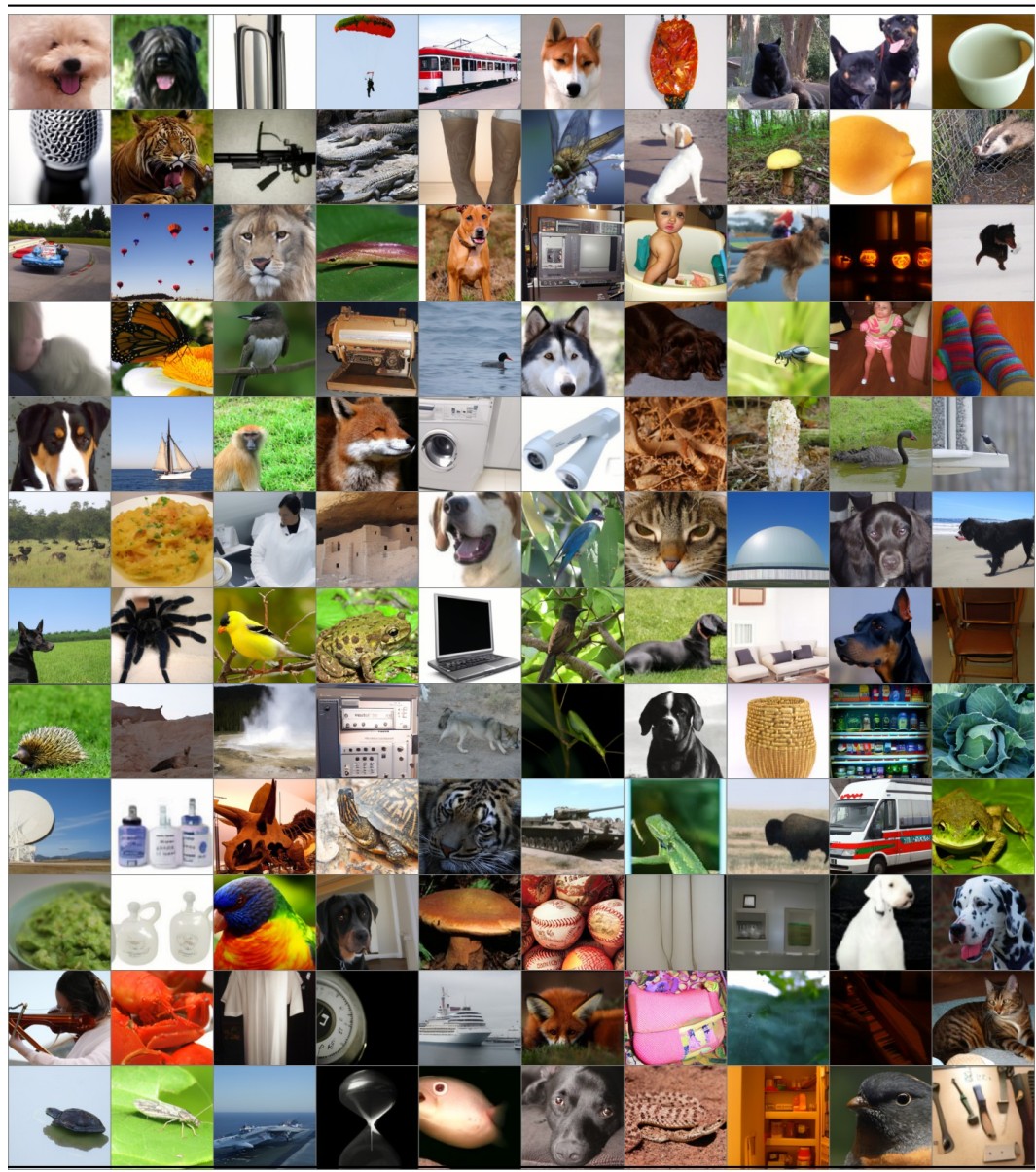

Figure 27: Random samples from our *RDM* , with $m = 0.01$ and classifier-free guidance with $s = 2.0$. Samples were generated with 100 DDIM steps.

ImageNet-Dogs

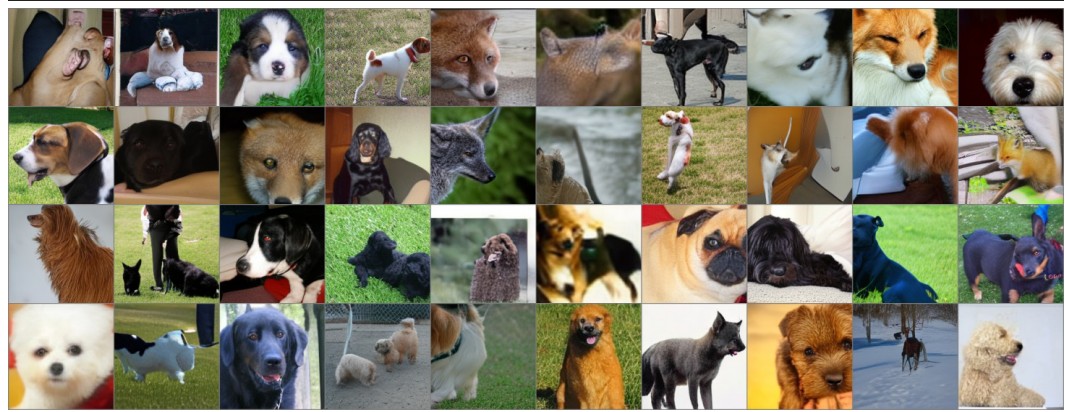

ImageNet-Mammals

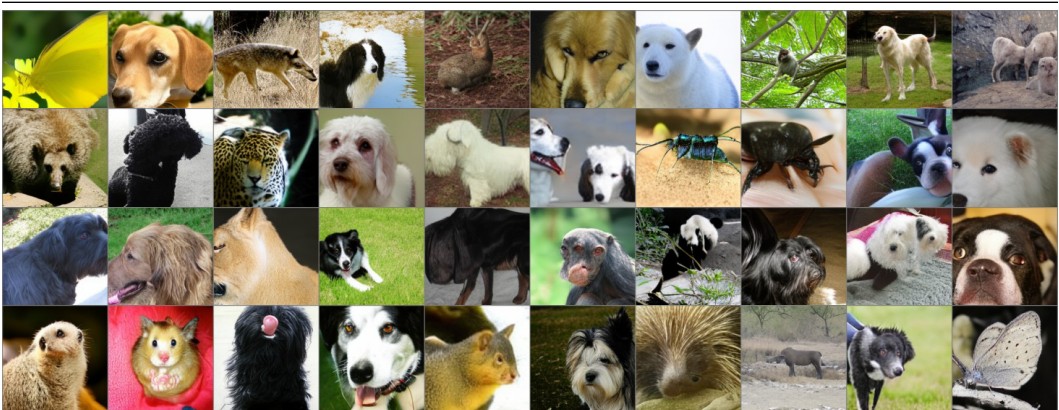

ImageNet-Animals

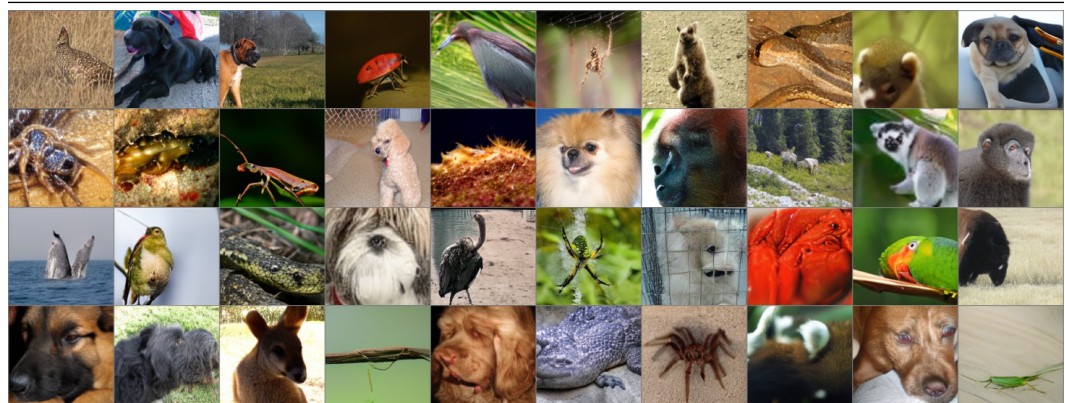

Figure 28: Random samples from our autoregressive models, with $m = 0.01$ and classifier-free guidance with $s = 2.0$. The models are trained on the dogs subset (top rows), mammals subset (middle rows), and animal subset (bottom rows). Samples were generated with top-$k = 4096$.

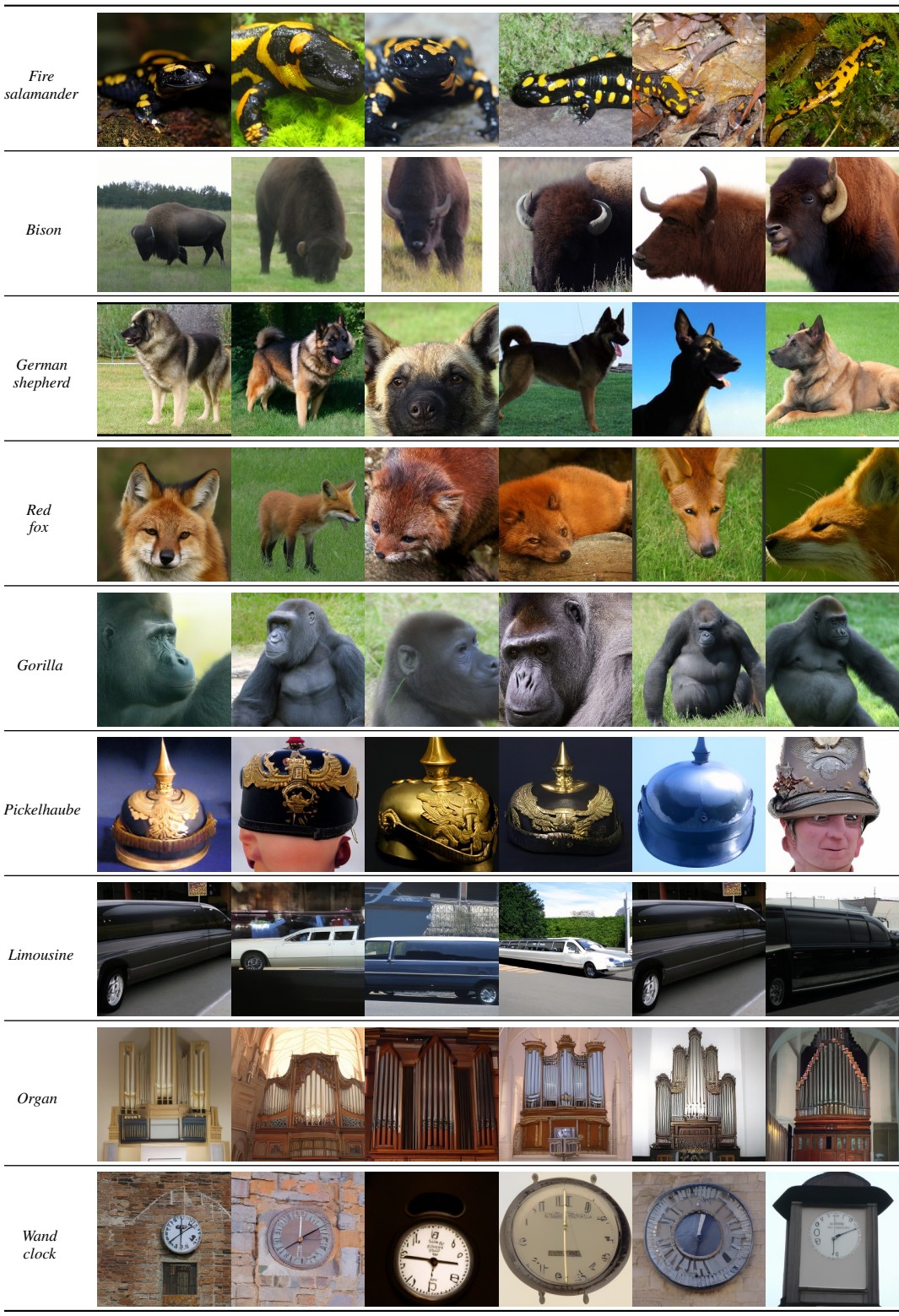

Figure 29: Additional class conditional samples obtained via the conditioning method presented in Sec. 3.3. Samples are generated with classifier-free scale $s = 2.0$ and 100 DDIM steps.

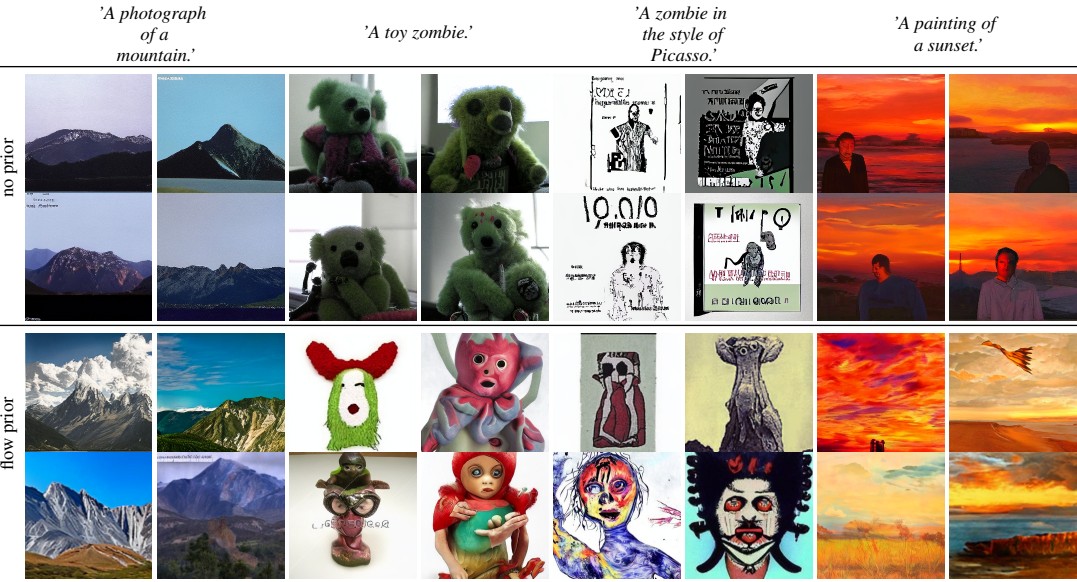

Figure 30: Text-to-image generalization in CLIP latent space needs a generative prior or retrieval in order to render diverse and high-quality images. Using the CLIP text embeddings directly produces flat, non-diverse samples, whereas the normalizing flow prior clearly improves quality and diversity. See Sec. 4.2

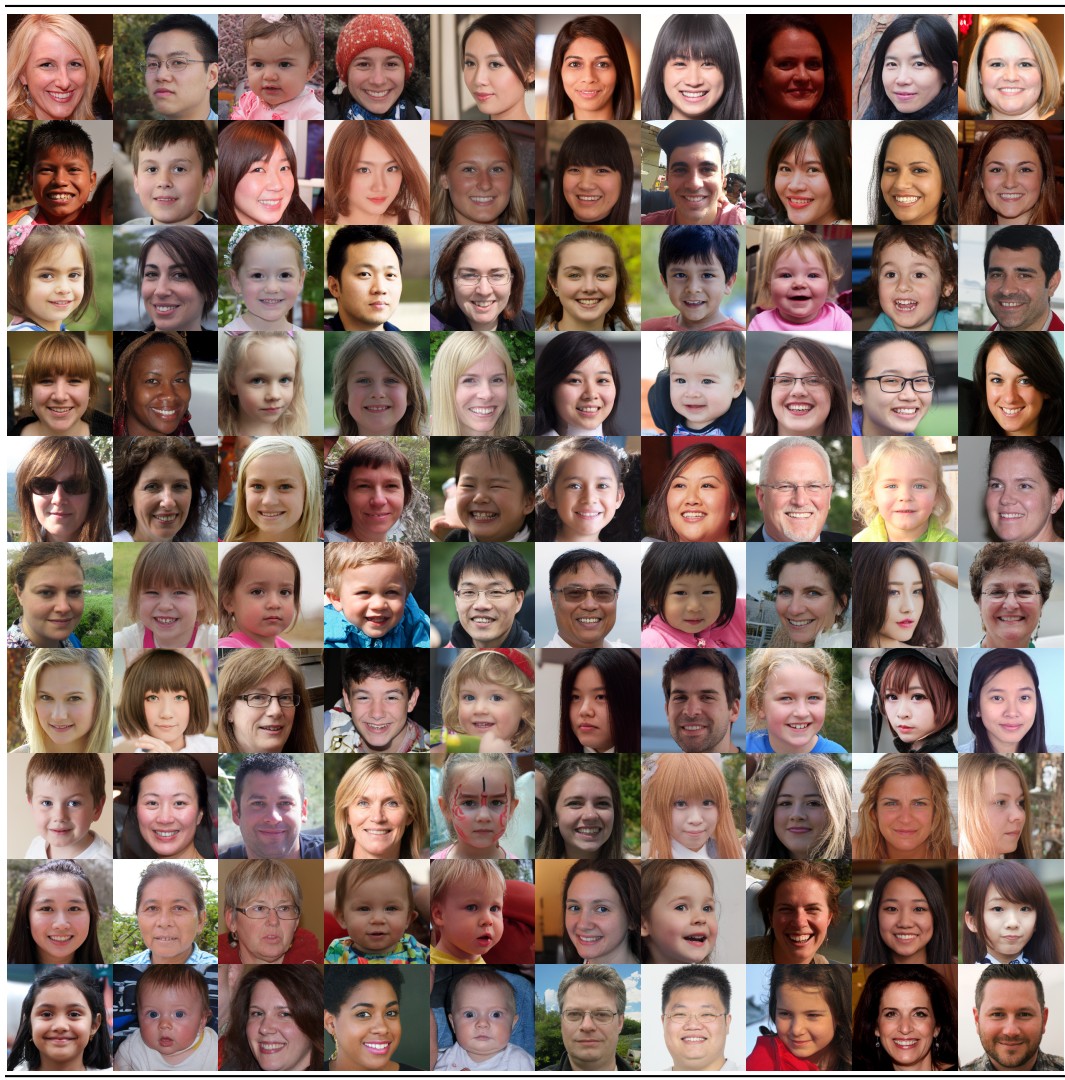

Figure 31: Random samples from our FFHQ RDM samples with 100 steps and $m = 0.01$.