# OpenReview forum: "Retrieval-Augmented Diffusion Models"
_NeurIPS.cc/2022/Conference — NeurIPS 2022 Accept_

### Official Review · Reviewer_XT8s · 2022-07-09

**Rating:** 6
**Confidence:** 4
**Soundness:** 4 excellent
**Presentation:** 3 good
**Contribution:** 3 good

**Summary:**

This paper proposes a method of using an external database of images to conditional a smaller generative image model for neural synthesis. Since the model is conditioned on the external database, some amount of domain transfer can be granted through changing the exemplars in the database. This external database also tackles a scalability problem, as retrieving nearest neighbor exemplars from the database is more efficient than trying to parameterize larger and larger datasets into a generative model.

Post Comments: Given the rebuttal of the authors addressing my concerns and an improvement of the ethics statement in the paper, I feel it's appropriate to increase my overall rating. I feel many of the timings per exemplar are a bit misleading, given that these are taken over large batched and amortized.

**Questions:**

1. Figure 1 is very misleading. There is a missing graph discontinuity on the X axis for model params (e.g. 0 is at 325 M params). Visually this is showing that the semi parametric models are around 1/8th the size when in fact they are 2/3rds to ~5/6ths the size. Additionally, this graph shows a flattening of FID as the database increases. Is this true of a model whose |theta| + |D| is the same size as the ADM w/classifier baseline?

2. L56, retrieving the exemplars at inference time is 0.95 ms. Since these images also need to be CLIP encoded, what is the time or compute spent to encode |D|.

3. One of the claims of this paper is domain transfer being possible with a change of the image database. Where any visually distinct domain transfers experimented with (i.e. style transfer).

4. What does the distribution of training data look like for parametric and semi-parametric approaches? Do semi-parametric models need less training time or less examples?

**Limitations:**

As in all generative work, there is the danger of generating images that can be harmful (used to incite fear, riots, harassment). The authors mention no limitations or negative societal impact of their work which I believe is an oversight. I don't think this research has any increased risk over other generative works, in fact, with careful curation of the database, this may actually *decrease* the potential to generate harmful content. The authors should more carefully consider the societal implications of their work.

**Strengths And Weaknesses:**

Strengths:
Use of an external database and retrieving nearest neighbor exemplars. Additionally, a database of images may not even be needed if you are pre-computing CLIP embeddings. This allows for a very efficient 2048 bits of representation per image.

With increasingly complex datasets, the semi-parametric models increase in recall performance (which is not true of the fully parametric baselines).

Weaknesses:
Domain transfer isn't explicitly shown. For example, a database of natural images used for training and cartoon images for exemplars to generate a "cartoon tiger" or "animated bear" would show a domain transfer from natural to animation images. The domain transfer examples shown are essentially the same visual domain as the training and the exemplars.

FID isn't state of art compared to classifier guided ADM, even with about 80% of the parameters.

---

> ### Author Response · Authors · 2022-08-02
> **Response to Reviewer XT8s Part 1**
>
> __Figure 1.__
>
> Thank you for your feedback on Figure 1. We have added an explicit visual marker to make it clear that the abscissa does not start at a value of 0. Additionally, we have replaced the orange markers with a line, since this is the same model for all choices of databases. The new figure can be seen here: https://imgur.com/LVmuxUl
>
> However, the course of the curve approaching the ADM value is rather coincidentally so. Other sampling parameters/guidance scales may give a potentially different picture, and models with different values for $| \theta|$ and $| \mathcal{D}|$ also behave differently --- ADM-G is in fact a different model that needs labels to train the guidance classifier. Our model does not need labels at any point for training.
>
> __Time needed to encode $\mathcal{D}$.__
>
> Using a batch-size of 100, our encoding pipeline takes 43 mins to process 1M examples from OpenImages (which corresponds to an encoding rate of ~385 samples/sec) while utilizing a single NVIDIA Quadro RTX 6000 GPU. The main bottleneck here is fetching data from hard disk, which is particularly expensive for OpenImages, since it contains large images up to 4K. Thus, with optimized dataloading (e.g. using file-streams as in HDF5/Webdataset) the pipeline could be even further speed up. However, since encoding a sufficiently large train database is a one-time effort that can be performed before the actual training, and since the extracted database can be used to train many models, we didn't implement advanced dataloading techniques. The encoded database can then be interpreted as part of the weights for a trained model and also used for inference.  A single forward pass through CLIP (ViT-B/32) takes 43ms/batch or 0.43ms/sample on a NVIDIA RTX 2080 Ti with batch size 100.
>
> __Visually distinct domains.__
>
> We would like to draw the reviewer's attention to Sec. 4.3 and Fig. 10 as well as Fig. 14 and 23 in the appendix, where we perform domain transfer between natural images and artistic images (via the WikiArt database). However, we also believe that this is an interesting point that deserves further investigation, and add experiments with the ArtBench database (cf. https://imgur.com/BQX0Blu for qualitative samples) that show that the model is able to perform transfers with different visual/artistic styles. In addition to that we also explore the transfer with the Pacs cartoon dataset [1]. In this dataset images are heavly stylized and have particularly distorted subjects compared to natural images. Images can be found here: https://imgur.com/cSuuiDM . We find that our model is able to stylize the images to some degree, however, it is not matching the style of the cartoon images perfectly. The model also struggles particularly when given the task so synthesize stylized giraffes (bottom right), a reason for this might be that the model has not seen any giraffes during training on ImageNet data.
>
> [1] Li, Da, et al. "Deeper, broader and artier domain generalization." __Proceedings of the IEEE international conference on computer vision__. 2017.
>
> __Training time for semi-parametric models.__
>
> Yes, our semi-parametric approach to image synthesis requires less training time to achieve the same quality as it "classic" counterparts. More precisely, we have added https://imgur.com/Nv9hWtp, which shows how FID, IS, Precision, and Recall for three instances of RDM (with different train databases, for a detailed explanation of the different instances of RDMs see Response to B1f9, Part 1, section 'Ablation studies on D' ) and LDM (RDM: 400M trainable parameters, LDM: 576M parameters) behave over progressive training on the Dogs subset of ImageNet. We see that the RDM achieves an FID of 50 about 3 times faster, while the recall (diversity) is consistently better than that of the non-retrieval model.
>
> __Results not SOTA compared to classifier guided ADM__
>
> We would like to point out a fundamental difference in these approaches: ADM-G __must__ train a classifier, because they __cannot__ use classifier free guidance as their model does not use any conditioning. In contrast, we condition on neighbors and can make use of various truncation techniques as introduced in l.199 (Sec. 3.3) and demonstrated in Sec. 4.5. Therefore, the results are not directly comparable and we solely included those to show that we can reach their performance even without access to labels.

---

> > ### Author Response · Authors · 2022-08-02
> > **Response to Reviewer XT8s Part 2**
> >
> > __Societal Impact & Limitations__:
> >
> > You are right, there is always the risk of severe abuse when the ability to generate artificial images is given to humans. We as researchers need to be aware of that issue and have discussed the societal impact in Appendix B, as well as limiations of our work in Appendix A.
> >
> > Thank you for the remark about the ability of our approach to curate the database to exclude potential harmful source images. When creating a public API that approach could offer a cheaper way to provide a safe model than retraining a model on a filtered subset of the training data or doing difficult prompt engineering. Conversely, this technique also allows for the inclusion of only malicious content, making it an easy way to create an explicitly toxic model.
> >
> > Another (more technical) limitation is an inherent tradeoff between database size (and associated storage and retrieval costs) and model performance, as evident from Figure 1. Storing and searching indices for databases of up to billions of images can become quite costly. Furthermore, our approach depends on the image representation that is chosen to encode images from $\mathcal{D}$ and the retrieval model. Both have significant influence on the performance of the RDM/RARM and further research needs to be done to determine the best choices here.

---

### Official Review · Reviewer_fYVT · 2022-07-11

**Rating:** 5
**Confidence:** 4
**Soundness:** 3 good
**Presentation:** 3 good
**Contribution:** 3 good

**Summary:**

This paper attempts to solve the problem of image synthesis (unconditional, conditional, text-guided) in a semi-parametric way. It extends IC-GAN [1] by introducing an external retrieval dataset (used for kNN search), and a pretrained fixed CLIP encoder (used for encoding image and text). The proposed method could be used on various image synthesis frameworks (e.g. Diffusion-based, Autoregressive-based models).

A fundamentally similar concurrent work is kNNDiffusion [2] (also mentioned by the authors in Appendix C)

[1] Arantxa Casanova, Marlène Careil, Jakob Verbeek, Michal Drozdzal, and Adriana Romero Soriano. Instance-conditioned gan. *Advances in Neural Information Processing Systems*, 34, 2021.

[2] Oron Ashual, Shelly Sheynin, Adam Polyak, Uriel Singer, Oran Gafni, Eliya Nachmani, and Yaniv Taigman. Knn-diffusion: Image generation via large-scale retrieval. *arXiv preprint arXiv:2204.02849*, 2022.

**Questions:**

Please see weaknesses above. Overall, I think this paper is a good extension of IC-GAN by introducing the retrieval-based methods to diffusion and autoregressive models, along with the combination with CLIP. However, some ablation studies, quantitative experiments, and more in-depth analysis with the proposed method are missing. If the authors could address these, I am willing to raise my score.

**Strengths And Weaknesses:**

Strengths

- The proposed method could be trained with images only, while allowing tasks including conditional, unconditional, text-guided image synthesis. This is achieved by aligning the latent space to a pretrained fixed CLIP encoder.
- Good experiments of introducing the idea of retrieval based image synthesis in diffusion models and autoregressive models.

Weaknesses

- One major difference the authors have mentioned against IC-GAN is the external retrieval dataset (as IC-GAN use the same retrieval and training dataset). How does changing retrieval dataset to a different one affect the results? Does the diversity matters or is it the dataset size matters? If we use the same training and retrieval dataset in this work, does the result differ a lot?
- Domain transfer by replacing the retrieval dataset is also presented in IC-GAN, what is the fundamental difference here?
- It seems that the quantitative experiments of text-to-image task on COCO, and class conditional synthesis on ImageNet against SOTAs are missing.
- In Figure 2, using text representation only performs better than using kNN samples and the combination of these two. Does this mean NN is not important for text-to-image synthesis? Then, what is the fundamental reason to introduce NN retrieval here.

---

> ### Author Response · Authors · 2022-08-02
> **Response to Reviewer fYVT**
>
> __Is NN not important for text-to-image synthesis?__
>
> Thank you very much for this question, which touches on a central point of our paper and which we would therefore like to try to present more clearly in our general response, see __"Neighbor-based approaches vs. paired data"__.
>
> __Difference between IC-GAN and our model for style transfer:__
>
> The main difference lies in the fact that during training (and inference) RDM uses feature encoding of multiple neighbor images instead of single instances. We believe that this increases the robustness of our model during test time to changes in the database. To support this claim, we have run the following evaluations: We use WikiArt as the inference database to generate 50,000 samples for both models. With these samples we evaluate FID, CLIP-FID, precision and recall of samples against the WikiArt dataset. Especially FID and precision demonstrate that our samples better approximate the WikiArt data manifold.
>
> |      **Method**     |      FID /CLIP-FID   |        Precision       |       Recall           |
> ----------------------|-------------------|-------------|-----------------|
> |           ICGAN         |    24.75/35.17  |              0.465       |     0.276       |
> |           Ours             |   21.50/13.01   |              0.628       |     0.336       |
>
> In addition, we also want to explicitly measure the transfer that happens when we exchange the database. To do this, we train a linear-probe on ResNet-50 features to distinguish images from ImageNet and WikiArt, our resulting classifier has an accuracy of 96% on an unseen validation-set. To see how well the method models the new database, we evaluate how many samples for a given inference dataset are correctly classified (i.e. D_train as ImageNet and WikiArt as WikiArt), see here https://imgur.com/0WBoVlm . We find that RDM produces images that the classifier identifies as better fitting for the given dataset.
>
> Of course a qualitative side-by-side comparison is also always important when evaluating generative models. For this we additionally take highly stylized images from the Pacs Cartoon dataset [1] as database to see the transfer capabilities in this extreme case. These samples can be found here: https://imgur.com/cSuuiDM . In addition to that there are more WikiArt-stylized images in Fig. 23 of our appendix.
>
> [1] Li, Da, et al. "Deeper, broader and artier domain generalization." __Proceedings of the IEEE international conference on computer vision__. 2017.
>
> __MS-COCO zero-shot FID__:
>
> Thank you for pointing this out. To show the full capabilities of our approach we evaluated the ImageNet-RDM from Fig. 2 on zero-shot COCO, see Tab. 1 in the "__general response__" section.
>
> __ImageNet class-conditional synthesis__:
>
> As requested by the reviewer, we present class-conditional results on ImageNet by using the strategy introduced in Sec. 3.3 in the main paper: **FID: 13.10; IS: 36.66; Precision: 0.67; Recall: 0.33** . These results are generated with 100 DDIM steps and c.f.g scale 1.5.

---

### Official Review · Reviewer_aPkQ · 2022-07-11

**Rating:** 5
**Confidence:** 4
**Soundness:** 2 fair
**Presentation:** 3 good
**Contribution:** 2 fair

**Summary:**

The paper proposes retrieval-augmented generative model, which is a semi-parametric model conditioned on retrieved features.

**Questions:**

In Table 1, ADM and RDM perform sampling with different hyper-parameters. Some more results are suggested for fair comparison (e.g. ADM with 100 steps or RDM with 250 steps).

Inception Score on MS-COCO dataset for text-to-image generation is suggested to be reported.

**Ethics Review Area:**

["I don’t know"]

**Limitations:**

Yes

**Strengths And Weaknesses:**

Strengths:

The proposed method is straight-forward and effective, good performances are achieved with smaller model size;

The paper is well-written and easy to follow;

The use of pre-trained CLIP model enables image generation, text-to-image generation and class-conditioned generation; Good experiment results are obtained on image generation task;


Weaknesses:

Some related works [1, 2] which also perform generation conditioned on CLIP features should be discussed and compared. Specifically, [1] also retrieve image features based on CLIP multi-modal joint feature space, and train the model based on retrieved features. The only difference is that [1] trains a diffusion model in a latent space;

The text-to-image generation results on MS-COCO dataset is worse than related works;

Diversity of generation is influenced

[1]. KNN-Diffusion: Image Generation via Large-Scale Retrieval. Oron Ashual, Shelly Sheynin, Adam Polyak, Uriel Singer, Oran Gafni, Eliya Nachmani, Yaniv Taigman.

[2]. LAFITE: Towards Language-Free Training for Text-to-Image Generation. Yufan Zhou, Ruiyi Zhang, Changyou Chen, Chunyuan Li, Chris Tensmeyer, Tong Yu, Jiuxiang Gu, Jinhui Xu, Tong Sun.

---

> ### Author Response · Authors · 2022-08-02
> **Response to Reviewer aPkQ**
>
> __Related works should be mentioned  and compared__
>
> kNN-Diffusion [1] is a concurrent work which we discuss in Appendix C. Unfortunately, no code has been published until now and we hence cannot compare it with this work more explicitly. We agree with the reviewer that comparing to LAFITE [2] improves our work. Therefore, we assess the common zero-shot metrics on COCO, which demonstrate that our model outperforms LAFITE in FID and achieves only slightly lower Inception Scores, despite being trained on a significantly smaller dataset . Please see the next question for the results.
>
> __Text-to-image worse than related work__
>
> We would like to point out that Fig. 7 contains results that were produced for an ablation study and not tailored towards beating the state of the art (evaluated on 2000 captions only, shallow models on ImageNet). To show the full capabilities of our approach we evaluated the ImageNet-RDM from Fig. 2 on zero-shot COCO, see Tab. 1 in the "__general response__" section. Here, we improve over LAFITE in FID although we train only on ImageNet, which has less than half of the size of CC3M, which is the train set for LAFITE, and our method does not require text prompts during training, whereas theirs does.
>
> __Diversity of generation is influenced__
>
> Yes, diversity is increased, as can be seen from our comparison in Fig. 11, where we compare the retrieval-augmented models to their fully parametric counterparts. As evident from the plots, both precision and recall are higher for both the autoregressive and the diffusion models. Further, FID is consistently lower, which is better. We therefore conclude that our retrieval-based approach improves diversity of generations while increasing visual fidelity.
>
> __Same number of sampling steps ADM vs RDM__
>
> For ADM we report the official values from the publication generated with 250 steps. Generally, for diffusion-based models, more sampling steps generally lead to better quality, so ADM is at an advantage. To have an absolutely fair comparison, we follow the suggestion of the reviewer and generate results with 250 sampling steps (Top-M 0.1, c.f.g scale 2.0) and achieved the following results **FID: 12.03 ;  IS: 79.78; Precision: 0.76, Recall: 0.55**

---

### Official Review · Reviewer_B1f9 · 2022-07-11

**Rating:** 6
**Confidence:** 4
**Soundness:** 3 good
**Presentation:** 2 fair
**Contribution:** 3 good

**Summary:**

This paper tries to use an additional database D to do generation. The idea is very simple and straightforward. During training for a training image x, they use it to find the k images from the D by using CLIP image encoder and K nearest neighbor. Then they feed those retrieved image CLIP embeddings into their generative model as extra information.

Because of the shared space between text and image, once their model is trained they can do text2image generation and class conditional generation even though the model is trained unconditionally.

They studied what is the best value of k, and propose a top-m sampling strategy for a trade-off between diversity and fidelity.




**Questions:**

Here are my major concerns for this paper regarding the setting and high-level:

1, The main claim of this paper is having an additional database. But, they do not justify why they want to have a disjoint D? What if D is the same as the training data? They should have shown what are the benefits of using different images as D. They have something similar in L289, but in this case k=1. They should properly study a model with k=4 when D is the same as training images.

2, They do not show enough ablation studies on the database. For example, how would different databases influence the model's performance and behavior? Although they studied the patch size used in OpenImage in the supp, those database features are all from the OpenImage, thus still similar in my mind.

3,  Similarly as above, they should also study the effect of the same database on the different training sets. They only conducted the training on ImageNet, what about the other dataset such as the single domain dataset (FFHQ, LSUN)? In these cases, how would the current database (OpenImage) influence the models? Does the database still provide useful information?  If not, why does someone need the database?

4, A lot of studies were conducted on text-to-image generation, zero-shot, etc.  But actually, the reason why they can do this is mainly because of CLIP, not because of the extra dataset. Thus I do think they present the paper appropriately. If they want to highlight these properties, then they should make their story based more on CLIP. If they still want to make their main story be the database, then they need to thoroughly study the concerns I listed above.

5, What is the true limitation of this paper? Although they mentioned limitations in the supp, but I do not think they take it seriously. What they mentioned are general problems for all diffusion models such as sampling speed. I hope to see they talk about the limitation unique to their approach.

Here are my major questions regarding the existing experiments and statements:

6, In table 1, they miss the LDM baseline.

7, In table 1, they should also list the database size, instead of only parameters. Since storage is also a main concern in a real-world application. They mentioned it in the footnote on page 4, but they should explicitly list it in table 1 to show the extra storage cost.

8, When talking about DALLE2, they said DALLE2 prior training requires paired data and they do not (L287), which I do not think is accurate. Since the CLIP training also needs paired data. Actually, I do not fully understand the purpose of L279-L295, if the goal of this part is to show their approach does not need to train a prior, then they need to show why a feature database is better than a prior trained on the features from the database.


Minor concerns:

1, in L157, I am not sure if LDM is the first one to propose cross attention. It seems GLIDE also uses cross attention to fuse image and text features.

2, for fig2, they can do a user study instead of only show 4 images.








**Limitations:**

My major question for this paper was they did not have enough study and analysis for training data and extra database, which is the key claim of their paper. Since they addressed them well in the rebuttal, I am happy to raise my score.

I still have the same concern about the way they present the paper. The main claim of this paper is using retrieval to help generation, BUT, the other very important component in their approach is CLIP which I do not think is replaceable. In other words, if I replace CLIP with normal Resnet trained on Imagenet, I do not think one can play around with text anymore. But I can still call my paper as "Semi-Parametric Neural Image Synthesis" or my model as "RDM". Thus I think their title and story is a bit over claim or not precise if one prefer.

In summary, the contribution of this work outweighs my concern. But I highly suggest to put their ablation on D and X conducted in the rebuttal to the main paper since these ablations backup their main claim. Also consider adding CLIP into the story when present the paper.

**Strengths And Weaknesses:**

Strengths:
Using an additional database to help generation. (I have a mixed feeling about this point as using an external bank of images is not a new philosophy in image generation)
The paper writing is very clear and easy to follow.
They achieve state-of-the-art unconditional generation results on ImageNet with less trainable parameters.

Weakness:

It lacks novelty. The model architecture follows LDM. Conditioning on CLIP embedding to enable the power of text2image generation is similar to DALLE2, and should be attributed to CLIP.

My main concerns/doubts are listed in the "Questions" section.

---

> ### Author Response · Authors · 2022-08-02
> **Response to Reviewer B1f9 Part 1**
>
> We here adress the individual concerns:
>
> * __It lacks novelty. The model architecture follows LDM. Conditioning on CLIP embedding to enable the power of text2image generation is similar to DALLE2, and should be attributed to CLIP.__
> This is a call for comparison with concurrent work published on ArXiv within two months (specifically April 13, 2022, see https://arxiv.org/abs/2204.06125) of the submission date for NeurIPS 2022 (which was May 19, 2022). Moreover, our method shows how to invert the multimodal space learned by CLIP without learning a specific text-image prior, and thus the statement that this "should be attributed to CLIP" is incorrect as written, since CLIP does not have generative text-to-image capabilities. Our approach does not require paired data for training the retrieval-augmented model and can be trained on images only, which is particularly interesting for high-performance multimodal contrastive models such as CLIP, since their training data is often not publicly available.  This is also related to the section __"Neighbor-based approaches vs. paired data"__  in our general response.
>
> * __Ablation studies on D__
> Thanks for this suggestion. Due to the fact that a larger and, thus, more diverse database should contain many smaller ones as its subsets, we skipped such a study in the original submission. But since the reviewer explicitly asked for this study, we provide this study in https://imgur.com/Nv9hWtp and compare the test performance of RDMs trained on the ImageNet-Dogs dataset with the dabases i) WikiArt (RDM-WA),  ii) COCO (RDM-COCO) and iii) OpenImages (RDM-OI) with an LDM baseline with 1.3 $\times$ more parameters. We see that choosing a database which does not provide the model with useful information for modelling the train data as in i) gives no improvement in FID and results in lower precision scores. However, using a small database containing useful information for the training task as in ii) results in lowered FID scores and higher recall compared to the LDM baseline. As expected, further increasing the database size as in iii) improves all metrics which we attribute to the statement made in the beginning of this answer. We expect the benefits of an increased database to become more prevalent for more complex datasets similar to Fig. 11 in the submission.
>
> *  __Missing LDM in Tab. 1; size of the database__
> We would like to point out that we show comparisons to LDM throughout the whole work, e.g. in Fig. 11, 19, and 20. As RDMs consistently achieve significantly better scores than the LDM baselines, and Fig. 11 additionally shows that using an external database is especially beneficial for increaslingly difficult datasets, we did not include LDM in the evaluation on full unconditional ImageNet. Nonetheless, as requested by the reviewer, we additionally trained an unconditional LDM on ImageNet with the same number of parameters and training setting (same hardware, batch size and number of train steps) as our RDM from Tab.1, which results in  **FID: 45.89; IS 23.46, Precision: 0.56, Recall: 0.55**. All of these results are worse than those of our RDM. We will add this to Tab. 1
> A database consisting of 20M visual instances takes 20GB storage space on hard disk when using the CLIP ViT-B/32 model and an FP16 representation of the encodings. Note, however, that we do not need to load this in the VRAM of the GPU and that we only use this for training. At test time, one can significantly decrease the storage requirements by using Top-M Sampling, see Figure 1.
>
> *  __Need to compare to prior and paired data__
> In l. 286-288 we explicitly state that "our retrieval-augmented approach provides an orthogonal approach to DALL-E 2 without requiring paired data" - training text-to-image generative priors as in DALL-E 2 requires, in contrast to our approach, paired data. See also our general repsonse, __"Neighbor-based approaches vs. paired data"__
>
> *  __I am not sure if LDM is the first one to propose cross attention. It seems GLIDE also uses cross attention to fuse image and text features.__
> GLIDE uses self-attention on the concatenated text and image representation sequences (vanilla attention complexity O((M+N)²), whereas LDM introduces conditioning via explicit cross-attention (vanilla attention complexity O(MN)).

---

> > ### Author Response · Authors · 2022-08-02
> > **Response to Reviewer B1f9 Part 2**
> >
> > * __Other datasets such as FFHQ, LSUN__
> > Here we provide additional samples on the FFHQ dataset using the OpenImages database as requested by the reviewer.  Since Kynkäänniemi et al [a] showed the standard FID to be 'very insensitive to the facial region', we use their proposed CLIP-based FID [a] to measure quality of samples. Here we outperform LDMs as well as strong GAN based models such as StyleGAN2 and Projected GAN. For LDMs we compare with two models for better comparability: The first LDM model is the officially released model from the github repository (https://github.com/CompVis/latent-diffusion) with the FID-optimal sampling parameters (250 steps, $\eta=1.0$). As our model has slightly more parameters (due to cross-attention for the conditioning) and is trained on 2 NVIDIA A100 with a larger batch size, we additionally train an LDM (same $N_{\text{params}}$) with the same number of parameters by using the same hardware and batch size. Thus, having a database is also beneficial for comparably easy, aligned datasets.   Samples can be found at https://imgur.com/OqrV2to
> >
> > | **Method** | CLIP-FID |
> > |:-----------|:------------:|
> > | Projected GAN | 4.87 |
> > | Style-GAN2 | 2.90 |
> > |  LDM  (from website)    |  2.12 |
> > | LDM (same $N_{\text{params}})$ | 2.63 |
> > | RDM | 1.92|  ==todo== |  ==todo==|
> >
> > [a] Kynkäänniemi, Tuomas & Karras, Tero & Aittala, Miika & Aila, Timo & Lehtinen, Jaakko. (2022). The Role of ImageNet Classes in Fr\'echet Inception Distance.

---

> > > ### Author Response · Authors · 2022-08-02
> > > **Response to Reviewer B1f9 Part 3**
> > >
> > > * __Societal Impact and "true limitations"__
> > > Although we are not sure which "true limitations" the reviewer refers to, we will add the following remarks to Appendix A, which already discusses some of the model's limitations.
> > >
> > > *Another limitation is an inherent tradeoff between database size (and associated storage and retrieval costs) and model performance, as evident from Fig. 1. Storing and searching indices for databases of up to billions of images can become quite costly. Furthermore, our approach depends on the image representation that is chosen to encode images from $\mathcal{D}$ and the retrieval model. Both have significant influence on the performance of the RDM/RARM and further research needs to be done to determine the best choices here.*
> > >
> > > *Furthermore, as pointed out by XT8s one should consider the ability to curate the database to exclude (or explicitly contain) potential harmful source images. When creating a public API that approach could offer a cheaper way to offer a safe model than retraining a model on a filtered subset of the training data or doing difficult prompt engineering. Conversely, including only harmful content is an easy way to build a toxic model.*

---

> > ### Comment · Reviewer_B1f9 · 2022-08-05
> > **size of the database and paired data**
> >
> >  Size of the database: Yes, I understand that one does not need to store the database into GPU memory during training, but besides RDM model itself, you still need to store features in disk for unconditional sampling or maybe other purposes (if I understand correctly). This is a concern in some real world applications such as deploying your approach into portable devices. Thus I still suggest to list the overall storage cost of your approach in inference time vs other approaches in inference time.
> >
> > Paired data: Thank you for the explanation. I understand that to solely train a RDM along, one does not need to have paired-data. But what I initially said is since you approach is built upon CLIP, which itself requires paired training data. In other words, the multimodal space is not free. If one replace CLIP with normal ViT or Resnet in your approach, I do not think RDM can still do text2img generation etc. Thus I feel "without requiring paired data" is a bit strong

---

> > > ### Author Response · Authors · 2022-08-08
> > > **Thanks for raising score**
> > >
> > > Thank you for raising the score, we are pleased to see that the reviewer is satisfied with our answers .
> > > Here are two further clarifications:
> > >
> > > **Size of database:**
> > >
> > > B1f9 is correct in that additional data is needed during sampling. However as shown in Fig. 1, our model is much more leightweight than ADM. Even with large $m=0.05$
> > >  (rightmost/bottommost dot in Fig. 1) our model has more than 50M parameters less than ADM (green diamond), while nearly bisecting FID score. Thus, it is actually better deployable than previous diffusion models, which in general have a higher parameter count than e.g. GANs. Thanks nonetheless for the suggestion. We will also add this information to Tab. 1.
> > >
> > > **Paired Data:**
> > >
> > > We agree that training CLIP required paired data. However, when we say "without requiring paired data", we mean that our model itself is not trained on paired data. The important aspect of this is that there is no need to gather a large scale paired-data set to further improve the model, as opposed to common text-to-image models as Dall-E2 and Imagen. This is -- as stated by B1f9 -- caused by the usage of the multi-modal model that was trained on paired data and enables us to use this shared latent space. But since we use a publicly available model (https://github.com/openai/CLIP), there would be no need to retrain CLIP to achieve our results. However, we want to emphasize, that our work does not solely make use of the multimodal space of CLIP and, thus, can also be realized with alternative pre-trained encoders $\phi$ which do not require paired training data. As shown in Section 4.3 and in this link (https://imgur.com/BQX0Blu), we can change properties of RDMs post-training only by replacing the train database with another one. This is a feat which is independent of the choice of $\phi$ and so are our proposed sampling strategies which lead to very flexible sampling behavior as shown in Fig. 12 and 22. Moreover, we would like to draw the reviewers attention to Fig. 19, where we see that RDM outperform LDMs with more trainable parameters in quality (FID, Precision) as well as diversity (Recall) not only with CLIP but also with pre-trained VQGAN as encoder $\phi$.

---

> > > > ### Author Response · Authors · 2022-08-09
> > > > **We will include the additiional ablations in the final version**
> > > >
> > > > As suggested by B19f we will include the ablations on D and X in the final version, just as all the other additional experiments presented here.

---

### Author Response · Authors · 2022-08-02
**General Response (Part 1)**

**General Response**

We thank all reviewers for their time and helpful comments, which we believe add value to the work. This general response aims to address questions shared by at least two reviewers; however, we also address each comment in separate responses.
To fully address some of your questions and remarks we want to add additional figures. All images that we have linked in all our replies can be found in this gallery: https://imgur.com/a/MoFrS5S

__Training on $\mathcal{D} = \mathcal{X}$ (retrieval database is training data)__

- B1f9 and fYVT asked about the benefit of having a retrieval database that is disjoint from the training dataset. This is an interesting question and to asses this we have used the rebuttal period to train another model on ImageNet that uses ImageNet itself as the retrieval database.
- In this experiment we found that sharing train set and database increases the visual quality (FID and Precision) on the training set. However, having a disjoint database increases the recall (diversity) of the samples as well as improves the generalization to new conditioning, e.g. COCO zero-shot text-to-image in form of FID and CLIP similarity. We believe this is caused by the model learning a more diverse set of neighbor features during training, which increases the robustness of the feature decoding. To further back this claim we conduct a second generalization study and replace the database during inference: For the model trained with the ImageNet database we use OpenImages at test time. Similarly, we use ImageNet as a test-time database for the model trained with the OpenImages-based database  RDM-IN/OI refers to the former,  RDM-OI/IN to the latter model. For RDM-IN/OI we see clearly deteriorated results, whereas RDM-OI/IN generalizes to the new database and even improves its performance compared to the standard setting, i.e. when using the train time database also during inference.  See Table 1 for the results.

 __Training data = $\mathcal{D}$__:

| Method |   FID  (train) | FID (val) |   CLIP-FID (train) |     CLIP-FID (val)    |     Precision train   |     Recall train     |
|------------| -----------------------------------------------------------------| -| -| -| -| - |
| **ImageNet** | -| -| -| -| -| - |
| RDM-IN        | 5,91      |  5.32   |    3.92     |  4.44  |   0.74        |       0.51       |
| RDM-OI        | 12.28   | 11.31    |     4.09     |  4.59 |    0.69      |       0.55      |
| **ImageNet with interchanged database** | -| -| -| -| -| - |
| RDM-IN/OI | 17.23  | 16.82 | 8.86 |9.75  | 0.52 | 0.60 |
| RDM-OI/IN | 10.81 | 12.01 | 3.84 | 4.41 | 0.81 | 0.39 |


| **Method** |   FID  | CLIP-FID |                CLIP Score             |     IS     |
| ---------------| ---------| -------------| -----------------------------------------| ----- |
|      **Coco zero-shot text2img** | | | | |
| LAFITE   |    26.94		|         n/a    |     n/a    |    26.02  |
| RDM-IN |    27.28             |     18.12     |    0.29    |  24.17  |
| RDM-OI |          22.08   |     13.16    |    0.30    |  24.31  |

---

> ### Author Response · Authors · 2022-08-02
> **General Response (Part 2)**
>
> __Neighbor-based approaches vs. paired data__
>
> B1f9 asked about the purpose of lines l.279--295, while fYVT wanted to know if the results in Fig. 2 imply that neighbors are not actually needed for our model. We try to clarify these questions in the following paragraph.
>  - The goal of models such as CLIP is to provide a shared space between images and captions. However, since there are always is a difference between images and their captions, e.g. images contain very specific information whereas captions may describe many different images. If one wants to use the utilize the similarity between both approaches this disconnected needs to be bridged. The need for this is demonstrated in Fig. 8: When a model is trained to reconstruct an image from its CLIP-embedding it fails when trying to generate an image based on a caption-embedding.
> - Other papers such as DALL-E 2 or LAFITE do this by training a transfer function between these two spaces, which requires paired data. We verify this by training an additional generative prior (a conditional normalizing flow) in Fig. 8 (note: we train this prior only for illustrative purposes and use paired data during training (a subset of the LAION 400M dataset)).
> - Our approach on the other hand works by providing the model with a more diffuse CLIP-image-embedding, i.e. by training the model with multiple embeddings in the form of nearest neighbors it is able to extract the high level information that is shared between all neighbors, making the model learn robust features that are also contained in captions. Therefore, retrieving multiple neighbors is a central contributor to the generalization of our model. This robust conditioning mechanism can then be used as in Fig. 2 to generate images from captions without having multiple neighbors at inference.
> - In summary, our approach provides an orthogonal approach to training a text-conditional image prior and does not require paired data. This is particularly interesting for open-source high-performance, multimodal models such as CLIP, since their training data is often not publicly available.

---

### Meta-Review · Area_Chair_ZgNJ · 2022-08-30

**Recommendation:** Accept
**Confidence:** Certain

**Metareview:**

This paper tackles the general image synthesis problem (unconditional, conditional, text-guided) using a semi-parametric manner. It first retrieves relevant samples from external dataset, and use them as additional conditions for image generation. It is verified with different image synthesis frameworks, e.g. Diffusion-based and Autoregressive-based models. The comprehensive experiments demonstrate the effectiveness of the proposed semi-parametric image generation methods, compared with baselines.

The paper received all positive review rating scores after some discussions, leading to an ``Accept'' decision overall.

**Award:**

No

---

### Decision · Program_Chairs · 2022-09-14

Accept